# A methodology to conduct wind damage field surveys for high impact weather events of convective origin

Oriol Rodríguez[1], Joan Bech[1], Juan de Dios Soriano[2], Delia Gutiérrez[2], and Salvador Castán[3]

[1]Department of Applied Physics - Meteorology, University of Barcelona, Barcelona, 08028, Spain
[2]Agencia Estatal de Meteorología, Sevilla, 41092, Spain
[3]Agencia Pericial, Cornellà de Llobregat, 08940, Spain

**Correspondence:** Oriol Rodríguez (orodriguez@meteo.ub.edu)

**Abstract.** Post-event damage assessments are of paramount importance to document the effects of high-impact weather-related events such as floods or strong wind events. Moreover, evaluating the damage and characterizing its extent and intensity can be essential for further analysis such as completing a diagnostic meteorological case study. This paper presents a methodology to perform field surveys of damage caused by strong winds of convective origin (i.e., tornado, downburst and straight-line winds). It is based on previous studies and also on 136 fieldworks performed by the authors in Spain between 2004 and 2018. The methodology includes the collection of pictures and records of damage on man-made structures and on vegetation during the affected area in-situ visit, as well as of available Automatic Weather Station data, witness reports and images of the phenomenon, such as funnel cloud pictures, taken by casual observers. To synthesize the gathered data, three final deliverables are proposed: (i). A standardised text report of the analysed event; (ii). A table consisting of detailed geolocated information about each damage point and other relevant data, and (iii). A map or a KML file containing the previous information ready for graphical display and further analysis. This methodology has been applied by the authors in the past, sometimes only a few hours after the event occurrence and, in many occasions, when the type of convective phenomenon was uncertain. In those uncertain cases, the information resulting from this methodology contributed effectively to discern the phenomenon type thanks to the damage patterns analysis, particularly if no witness reports were available. The application of methodologies as the one presented here is necessary in order to build homogeneous and robust databases of severe weather cases and high impact weather events.

## 1 Introduction

Meteorological phenomena associated with strong surface wind of convective origin (i.e., tornadoes, downbursts, straight-line winds) can cause important disruption to socio-economic activity, including injuries or even fatalities despite their local character compared to larger scale mid-latitude synoptic windstorms or tropical storms. For example, from 1950 to 2015, tornadoes in Europe caused 4,462 injuries, 316 fatalities and economic losses of at least €1 billion (Antonescu et al., 2016, 2017). Due to their economic and social impact, a large number of works have been devoted to the study of these phenomena both from a meteorological point of view (e.g., Taszarek et al., 2017; Miller and Mote, 2018 or Rodríguez and Bech, 2018) or from the point of view of their consequences (e.g., Rosencrans and Ashley, 2015 or Strader et al., 2015).

The systematic elaboration of post-event forensic field surveys is still the standard way to evaluate the damage caused by a strong-convective wind event (Marshall, 2002; Marshall et al., 2012; Zanini et al., 2017), despite the recent progress on assessing wind damage using remote sensing data such as high resolution radar observations (Wurman et al., 2013; Wakimoto et al., 2018). A detailed damage analysis from these meteorological phenomena allows to estimate the wind intensity using a wind damage scale such as the Fujita scale (F-scale, Fujita, 1981) or the Enhanced Fujita scale (EF-scale, WSEC, 2006). Similarly to field surveys of hailstorms (Farnell et al., 2009) or floods (Molinari et al., 2014; Li et al., 2018), wind damage fieldworks contribute to a better characterization of the affected area, making possible to estimate the length and width of the damage swath (e.g., Burgess et al., 2014; Meng and Yao, 2014; Bech et al., 2015). Moreover, in-situ damage surveys are especially useful to determine which phenomenon took place when there is an absence of observations by analysing damage patterns on forest and how debris is spread (Hall and Brewer, 1959; Holland et al., 2006; Bech et al., 2009; Beck and Dotzek, 2010; Rhee and Lombardo, 2018). This information can be added to natural hazards databases such as the USA Storm Prediction Center Severe Weather Database (Verbout et al., 2006) or the European Severe Weather Database (Dotzek et al., 2009), making possible building up robust and homogeneous datasets, improving the knowledge of spatial-temporal distribution and characteristics of tornadoes, downbursts and straight-line winds.

Currently, fieldworks are usually performed to assess damage of specific strong-convective wind events (e.g., Lemon et al., 2003; Bech et al., 2011; Wesolek and Mahieu, 2011), but rarely to analyse in detail most of the reported cases. The timing and economical costs, especially when helicopter flights are used, prevents to carry out in-situ damage analysis frequently (Edwards, 2020), particularly out of the USA. Therefore, there is a need for a methodology to conduct wind damage field surveys for high impact weather events of convective origin easily reproducible anywhere, which should be efficient to optimize time and economic resources allowing the study of as much reported events as possible.

The objective of this paper is to propose a methodology to conduct in-situ damage surveys of strong wind events from convective origin. It can contribute to improve the detection, mapping and characterization of wind damage in a homogeneous way, which is important to better describe specific meteorological phenomena, with the particularities associated with damage from convective local storms. Therefore, the main goal of the proposed methodology is to gather as much geo-referenced information (pictures and records) as possible about relevant damaged elements (i.e. man-made structures and vegetation) to reproduce the damage scenario. This information should be complemented with other available data, such as witness enquiries, data from Automatic Weather Stations (AWS) located close to the affected area, remote-sensing data, and images of the phenomenon together with their location and orientation to analyse strong-convective winds phenomena from a meteorological point of view.

The methodology presented here is based on previous studies (McDonald, 1984; Bunting and Smith, 1993; Gayà, 2018; Holzer et al., 2018) and also on 136 wind damage surveys performed between 2004 and 2018 by the authors. All the analysed events have been recorded in Spain (south-western Europe), which includes a vast majority of the Iberian Peninsula and also the Balearic and Canary Islands. Nevertheless, most of these fieldworks have been carried out in Catalonia and Andalusia regions (Fig. 1), where high-densely populated areas are frequently affected by tornadoes (Bech et al., 2007, 2011; Mateo et al., 2009; Sánchez-Laulhé, 2009; Gayà et al., 2011; Riesco et al., 2015). Three final deliverables are suggested to synthesize

the data recorded: (i). A text report of the analysed event; (ii). A table consisting of detailed geolocated information and, iii). A map or a KML (Keyhole Markup Language) file containing the previous information ready for graphical display and further analysis.

The rest of the article is organized as follows. Firstly, in Sect. 2 it is provided an overview of previous in-situ fieldwork techniques. Section 3 describes in detail the field survey methodology proposed. In Sect. 4 specific strengths and limitations of the methodology are discussed, as well as possible uses of fieldwork data. Finally, Sect. 5 presents a summary and final conclusions of the study. As Supplementary material, an example of deliverables (text report, table and KML file) of a damage survey of a recent tornadic event is provided with the aim to better illustrate the methodology proposed and to facilitate its application.

## 2 BACKGROUND

McDonald (1984) and Bunting and Smith (1993) provided guidelines to carry out strong-convective wind damage surveys. There, the process of mapping data by locating images taken during the fieldwork was challenging due to non-digital cameras and the absence of Global Navigation Satellite System incorporated on these devices. Both documents recommended to complement surface observations with aerial images if available and, in the second one, it was also explained how to treat with direct witnesses and propounded to ask them for specific information about the event and damage.

On the other hand, the analysis of historical events such as Gayà (2007) and Holzer et al. (2018) showed the utility of press references and in-situ images taken by witnesses on reconstructing tornado damage paths. They pointed out the necessity of geo-referencing the locations where photos were taken and the damaged elements, using GIS tools and triangulation methods. Furthermore, Holzer et al. (2018) provided useful indications for current fieldworks, such as visiting affected areas as soon as possible and also to provide an estimation of the wind intensity for each damaged element given by the pair Damage Indicator-Degree of Damage (DI-DoD) from the EF-scale (WSEC, 2006), similarly to other authors such as Burgess et al. (2014).

During the first decade of the current century, the use of GPS receivers or similar systems was extended on in-situ damage assessments to geolocate gathered data, as discussed in Edwards et al. (2013). Moreover, the use of aerial imagery from helicopter or airplane (e.g., Fujita, 1981; Bech et al., 2009) and high-resolution satellites (e.g., Molthan et al., 2014; Chernokulsky and Shikhov, 2018) has also been frequent to analyse damage swaths. Recently, drones have been raised as a new device which might be useful to, at least, complement surface surveys (Bai et al., 2017).

## 3 METHODOLOGY

The methodology to carry out damage surveys must be efficient, allowing to visit the affected area in the shortest time possible. It must be also easily reproducible, and its results should be accurate. Geolocating damage using pictures or videos recorded with smartphones, or cameras with a Global Navigation Satellite System such as GPS fulfils these conditions (Edwards et al., 2013). Nevertheless, as it happens with other types of damage assessments, there are inherent uncertainties that should be

taken into account when analysing field data (Beven et al., 2018), like possible GPS location errors or ambiguous application of intensity-rating assessments due to EF-scale limitations, which are discussed on Sect. 4.

In Table 1 are summarized the main devices needed to carry out fieldworks throughout the proposed methodology. Moreover, as indicated in Bunting and Smith (1993) and Gayà (2018), water, food, comfortable footwear, rain jacket, spare clothes and a mobile phone spare battery are recommended, because affected areas may be far away from inhabited locations. As surveyor displacements longer than a few kilometres can be required, a well-equipped, preferably all terrain, car is necessary to save time between points-of-damage analysis. Nevertheless, difficult access areas may be found along the track, because of muddy roads and fallen trees or simply because of the absence of roads. Especially in these cases, and also to study in detail damaged areas, walking is the basic way to perform the field survey.

Despite this may not be always feasible, it would be ideal that the damage survey team was multidisciplinary, being formed by meteorologists, insurance inspectors, forestry engineers and architects experienced on damage assessments, preferably familiar with damage reporting systems such as the EF-scale. This would facilitate an accurate and detailed analysis of the damage and the phenomenon intensity.

The proposed methodology is organized in three stages (Fig. 2). The first step includes pre in-situ damage survey tasks, preparing the actual visit of the damaged area (Sect. 3.1). Secondly, the in-situ fieldwork tasks, which include direct gathering of man-made structure and vegetation damage information, and also collection of direct witness experiences (Sect. 3.2). Finally, post in-situ damage assessment tasks, which involve the organisation of all the information collected into three deliverables (a text report of the event, a geolocated information table and a data location map; Sect. 3.3).

### 3.1 PRE IN-SITU SURVEY TASKS

To properly prepare the damage survey, a number of previous tasks must be performed. One of them is planning the route of the fieldwork. As mentioned in Holzer et al. (2018) it is strongly recommended to start in-situ damage surveys as soon as possible, especially if urban areas have been affected. Emergency and clearing services may start repairing tasks only a few hours after the event, which can alter the quality and quantity of possible information available during the fieldwork. Thus, to optimize time and resources, a detailed planning is necessary to carry out the in-situ damage assessment.

Firstly, preliminary information should be collected about damage location and images available on the media and social networks, which are the main providers of strong-convective winds reports nowadays (Hyvärinen and Saltikoff, 2010; Knox et al., 2013; Kryvasheyeu et al., 2016). Citizen science collaborative platforms covering different geographical domains such as the European Severe Weather Database (ESWD, Dotzek et al., 2009), the severe weather database of the Spanish Meteorological Agency (SINOBAS, Gutiérrez et al., 2015) or the meteorological spotters platform of the Meteorological Service of Catalonia (XOM, Ripoll et al., 2016), are also examples of valuable sources of tornado and downburst reports.

Contacting emergency services and local authorities can also provide valuable information, as they may record detailed damage data, especially if an urban area is affected. This kind of information may be crucial for post in-situ damage study because clearing services might start arrangement tasks before the in-situ visit is started. Occasionally, they may take damage aerial recordings, which can be very useful to complement the damage survey assessment.

To consider performing an in-situ damage survey, the report must contain information about damage and/or a well-developed funnel cloud (i.e., a funnel cloud extending down below cloud base at least 50 % the distance between the cloud base and the ground level). Funnel clouds (Fig. 3) are a typical feature of tornadic storms though sometimes may form without developing a tornado, i.e. when the air rotating column associated with the funnel does not reach the ground. When damage reports are available (Case 1 in Fig. 2), their location should be found by contacting their authors and/or using GIS cartography, proceeding

as the case described in Holzer et al. (2018). Applications such as Google Street View can be very useful to carry out this task.

      Nevertheless, if damage reports are not available but only developed funnel cloud images are reported (Case 2 in Fig. 2) then authors have to be asked for the location where photos were taken and their orientation. If for any reason this information is not accessible, it should be estimated from meteorological observations such as weather radar and satellite imagery (e.g., comparing radar images and the location of precipitation features observed in photos of the event respect to the funnel cloud,

as described by Wakimoto and Lew, 1993; Wakimoto and Liu, 1998; Zehnder et al., 2007) and GIS cartography. Then, from the triangulation of those pictures of the funnel cloud it is possible to preliminarily identify a possibly affected area, which can be more precisely delimited when the number of photos or videos from different perspectives is high (Rasmussen et al., 2003). However, at this stage it has to be kept in mind the possibility that the funnel cloud may have not produced damage, either due to the lack of man-made structures or trees in the area intercepted by the tornado, or because the strong rotation associated to

the funnel cloud actually did not touch down. This possibility will be verified during the in-situ survey tasks.

      Analysis of satellite and weather radar imagery is required to estimate the approximate timing of the event and the movement of the convective parent storm that may have produced the phenomenon. That information should be considered in order to extend the initial evidences of a preliminary damage path (looking for possible initial and ending damage path points) and to assess the consistence of reports by eyewitnesses.

On the other hand, existing AWS in the area of interest can play an important role to determine the phenomenon type, the timing of the event and also to estimate the wind strength (Letchford and Chay, 2002; Karstens et al., 2010). Therefore, it is strongly recommended to search and locate all weather stations in the area of study, requesting the data with the maximum temporal resolution, and performing basic quality control (time consistency and comparison with official observations) before use. High-temporal resolution wind data series during the passage of a tornado are usually characterized by a sudden increase

of wind speed and a swift direction shift, as it is shown in Figure 4. By contrast, although a downburst event is also described by a wind strengthening, there is a predominant wind direction without relevant changes (Orf et al., 2012).

      Another important task before starting the actual in-situ damage assessment is to check the wind climatology of the studied area, particularly in windy regions (either because of the orography or the prevailing synoptic conditions; Feuerstein et al., 2011). In this case, man-made structures and vegetation are adapted to resist strong winds –sometimes from specific directions–

and wind speed thresholds over which an element can be damaged may be higher than in non-windy regions. Therefore, if a weak tornado or microburst affects a region usually influenced by strong winds, it is possible that little or no damage is found. Similarly, the application of an intensity damage scale in very windy regions may require some adjustments –i.e. increasing the wind speed thresholds for specific damages– as discussed in Feuerstein et al. (2011).

In some occasions, the studied area may have been affected recently by another damaging windstorm or by a heavy snowfall

which may have produced widespread damage in forests – for example due to wet snow as described in Bech et al. (2013) and Llasat et al. (2014). In those cases, the data collection process may be hampered by possible overlapping damage and, consequently, great care must be taken to identify the most recent one. A possible way to mitigate this problem is asking locals about previous events and paying attention to the dryness from affected trees and broken branches, which can indicate if forest damage is recent or not.

## 3.2   IN-SITU SURVEY TASKS

To avoid alterations of the damage scenario due to clearing services, the fieldwork should preferably start on the most resilient areas, i.e. where socio-economic activity is more intense and are more likely to recover quickly. The proposed priority order is to visit urban areas first, then damaged electrical transmission or telecommunication lines, industrial parks and urban parks and, finally, forest and other surrounding areas (Fig. 2).

As a general principle, the highest possible number of relevant damaged elements should be analysed on the affected area, both man-made structures and natural (vegetation) elements. Moreover, if any previously unknown AWS is detected during the fieldwork, it should be considered to contact with its owner asking for data. The same process should be carried out for outdoor security cameras, which may record the event and could provide valuable information in order to determine which phenomenon took place. Interviews with eyewitnesses, which can provide key information about the event and other damaged

areas, are also very important.

All the in-situ measurements are related to geo-referenced damaged elements. To reduce the time of registering data in order to proceed to the visit of other affected areas, it is proposed to take a photo from the measuring device showing clearly the data (Fig. 5), following the order proposed in Table 2 (from V5 to V10). After that, a photo of the damaged element should be taken, whose metadata already contains latitude and longitude. Therefore, surveyors can associate each measure to each geolocated

element during the post in-situ tasks when organising the records gathered in the fieldwork.

Note also that Table 2 lists maximum uncertainties recommended for each measure type reflecting possible maximum errors in the field survey measures as suggested in Beven et al. (2018). Particular uncertainty values listed in Table 2 are consistent with the resolution of data presented in several severe weather databases such as NOAA/SPC (2019), ESWD (Dotzek et al., 2009), SINOBAS (Gutiérrez et al., KERAUNOS (2020), 2015) and Gayà (2018), where damage path width is usually expressed

with a resolution of $\pm 10$ m (i.e., damage location uncertainty must be smaller than $\pm 1 \times 10^{-4}$ deg.). Furthermore, uncertainties listed also take into account surveyors experience and the data resolution from previous studies – e.g., direction of fallen trees and wind-borne debris are typically presented with 5 deg. range (Bech et al., 2009, 2011, 2015).

### 3.2.1   MAN-MADE STRUCTURES DAMAGE ASSESSMENT

Man-made structural damage analysis is essential to estimate the phenomenon wind intensity, for example using the EF-scale.

As explained in WSEC (2006), the Enhanced Fujita scale considers several Degrees of Damage (DoD) from a total of 23

Damage Indicators (DI) related to constructions and 3 DI from other man-made structures that can be used to determine the 3-seconds wind gust speed associated to these damages.

In the present methodology it is proposed to geolocate every damaged structure on the affected area, whose coordinates (latitude and longitude) can be obtained from the GPS receiver of the photo camera (with a precision greater than $\pm 1 \times 10^{-4}$ deg., see Table 2). It is also convenient to take one or more pictures from each damaged element, both general and detailed views that may be of interest to evaluate the damage intensity (Marshall et al., 2012; Roueche and Prevatt, 2013). These photos should also be used during the post in-situ damage survey analysis to study which type of strong convective wind phenomenon caused the damage.

Moreover, for each affected man-made structure, the pair of DI-DoD data values should be provided by using an intensity-rating scale as the EF-scale, as proposed by several authors such as Burgess et al. (2014) and Holzer et al. (2018). This task can be carried out during the damage survey, but it is recommended to be performed during the post in-situ damage assessment analysis. The main reason is to optimize the time and sources devoted to the in-situ survey. In case that no DI could be associated to the damaged element, it should be explicitly shown as 'unrated'.

It is highly recommendable to check the maintenance status of the damaged man-made structures to avoid a biased intensity determination. Previous weaknesses or deficiencies on construction can make structures more vulnerable to strong winds and so a higher Degree of Damage might be caused for an expected wind speed (Doswell et al., 2009). For example, if an absence of anchors or the presence of rust on metal beams from a roof are observed, this should be explicitly documented by pictures and a brief description to be taken into account when a damage-rating scale is applied, as already proposed by Fujita (1992).

The estimated trajectory and distance covered by wind-borne debris, as well as its size and weight, may also provide valuable information to estimate wind velocity associated to the studied phenomenon (Knox et al., 2013). Therefore, it is recommended to measure the dragged or flying distance and direction of objects of interest, if origin and final locations are known, using a tape measure or GIS tools (Table 2). It is also interesting to document its weight, either estimated consulting bibliography or measuring it by a portable balance in case of small objects (the relative error should be less than 10 %).

### 3.2.2   FOREST DAMAGE ASSESSMENT

As mentioned in previous studies (see for example Holland et al., 2006 or Bech et al., 2009), the maximum wind field (direction and intensity) associated with a strong-convective wind event can be approximately derived from the fallen trees pattern. Therefore, if a substantial number of trees are damaged to produce a clear damage pattern, a detailed forest damage study is recommended. As described in detail in the Appendix, if fallen trees present a convergence and rotational pattern along a linear path, it is likely it was caused by a tornado, whereas if a divergent damage pattern, mostly nonlinear, is observed, the most likely cause is a downburst. This analysis is especially interesting for those cases where there is no image nor direct witness of the phenomenon to determine the damage origin.

The forest damage survey should be carried out similarly to the man-made structure damage assessment, taking pictures of every relevant damaged vegetation element and registering its location (latitude and longitude). In case of uprooted trees, the fall direction (azimuth) should be measured using a compass with, at least, 5 deg. of precision (see Table 2). However, it should

be noted that fall tree directions may be influenced by local factors and might not be representative of the wind direction. For example, trees falling on a steep slope terrain (favouring one fall direction over others) or the presence of another nearby tree falling first can alter the tree direction with respect to the dominant wind. Therefore, in these cases it is recommendable not to consider the data. In case of snapped trees, trunk diameters should be measured with a measuring tape (with a minimum resolution of 5 cm; Table 2). This data can help in the damage-rating task. However, as there may be a large number of damaged trees in a forest area, it is advisable to collect data from the most representative ones (for example, where tree fall direction changes or converges, probably indicating the effects of air rotation; or where damage is most significant, and surrounding damaged trees to delimitate the damage swath width).

Damage in forest areas can be also useful to evaluate the phenomenon intensity. The EF-scale (WSEC, 2006) describes different wind velocity ranges for five Degrees of Damage (DoD), namely small limbs broken, large branches broken, trees uprooted, trunks snapped, and trees debarked with only stubs of largest branches remaining. As wind effect on trees also depends on the tree species (Foster, 1988; Fig. 6a), the EF-scale also distinguishes between softwood and hardwood trees. Thus, DI-DoD pairs for each analysed vegetation element should be provided.

Moreover, soil characteristics can affect tree stability; in case of very moist soil, or thin soil over rocky subsoil, trees can be uprooted more easily, as it is illustrated in Fig. 6b. Trees health can also alter the resistance to strong winds. As it is done for man-made structures, these debilities must be stated in the report. In order to refine intensity-rating tasks in forests, it is recommended to calculate the ratio of affected trees in 50 m x 50 m areas if possible; it can be related to the EF-scale, according to Godfrey and Peterson (2017). High-resolution aerial imagery (i.e. from helicopter or drone) can be useful to carry out this task. This analysis is especially interesting in the most severely affected forest area of the damage swath.

Most tornado damage paths are less than 5 km long; for example, in Spain only 25 % of tornado identified tracks are longer than 5 km (Gayà, 2018). Therefore, a detailed forest damage analysis is usually possible. However, in cases where damage is widespread, a complete detailed analysis may not be feasible. In this case, it is recommended to study discontinuous segments every 250-500 metres along the expected damage swath. This allows estimating the path width and looking for the damage continuity. In addition, as previously commented, aerial images can enhance the forest damage analysis, especially in case of large damage tracks and difficult access areas (Karstens et al., 2013). Alternative approaches to surveys over widespread forest damaged areas are satellite image processing, as recently developed by Molthan et al. (2014), Chernokulsky and Shikhov (2018), Shikhov and Chernokulsky (2018) and Shikhov et al. (2019).

### 3.2.3 WITNESS ENQUIRIES

Direct witnesses, if available, are an important source of information often essential to determine which type of strong convective wind phenomenon occurred. Witnesses experience of the event and their possible knowledge of other casual witnesses in nearby damaged locations can be very useful to complement a damage survey. In Bunting and Smith (1993) and Gayà (2018) it is noted that a direct witness may have been emotionally or physically affected by the phenomenon (for example private property damaged or close persons injured) so it is necessary to be respectful and careful during the enquiry.

It is important to let witnesses explain in their own words their experience of the event, and interviewers should avoid using key words such as tornado, downburst or gust front, particularly in those cases when the phenomenon type is not known yet. The terms used by the witness may provide valuable clues about what happened. In addition, it is necessary to consider that previous media reports can alter the explanation of witnesses; for example, if the event has already been described as a tornado in the media, even if evidences of rotation are not found in the damaged area, people will probably say that a tornado has occurred.

A brief and concise inquiry, with specific questions but allowing open answers that may unveil relevant information, is proposed. Recommended questions are shown in Table 3. Moreover, in some occasions a direct witness may have taken photos or videos of the phenomenon that can be helpful for the study. When pictures are available, they should be treated as described in Sect. 3.1.

### 3.3 POST IN-SITU SURVEY TASKS AND DELIVERABLES

When the in-situ damage survey is completed, the event analysis should be complemented revising meteorological remote-sensing data, which now can be compared with the records obtained in the survey. The information collected by direct witnesses, pictures, and videos usually allow to restrict the event occurrence to a temporal window of about 15 minutes to 2 hours. Satellite imagery and data from Doppler radar, lightning detection systems and AWS (particularly if located within or close to the damage swath) from the period of interest can provide the necessary information to verify that the identification of the convective structure as the responsible of the damage performed during pre in-situ damage survey was correct. In particular, the starting and ending time of the event can be estimated by checking the time when the convective structure passed over the initial and the final point of the damage swath, respectively, with an error typically less than 5 minutes. It is recommended to perform this comparison with Doppler radar observations, if available, with data in original polar coordinates keeping the highest spatial resolution (see for example Bech et al., 2009, 2011, 2015). In some cases, it is even possible to estimate the mean translational velocity and direction of the convective cell, knowing the distances between initial and final damage path and the starting and ending time of the event. This can be very useful to compare theoretical surface wind vortex models with observed damage patterns over forest areas (Bech et al., 2009; see the Appendix for further details).

Finally, all the information gathered needs to be organized and archived in an easily interpretable way to analyse the strong-convective wind event. In the following subsections, three final deliverables are proposed to achieve this objective: (i). A standardised text report of the event, (ii). A geolocated information table, and (iii). A data location map. These deliverables are illustrated explicitly with the example of the 15 October 2018 Malgrat de Mar – Massanes tornado case (see location in Fig. 1), provided as Supplementary material. Then, according to the flow diagram shown in Fig. 2, in Case 1 and in Case 2 and new damage found, it can be concluded that a strong-convective damaging wind event (tornado, downburst, straight-line winds) occurred, whereas if a developed funnel cloud was reported and during the fieldwork no damage was found, it might be deduced that funnel cloud did not touch down or there was not any exposed and/or vulnerable element in the tornado track to be damaged.

### 3.3.1 TEXT REPORT OF THE EVENT

The text report of the event should be an overview of the analysed episode, including a brief description of the information gathered during the fieldwork and the main conclusions from the analysis of these data. The proposed deliverable is divided in seven parts.

• General event information. It includes geographic data of the analysed meteorological phenomenon following current international standards for disaster reports losses (De Groeve et al., 2014), such as names and codes of country (ISO 3166-1 alpha-3 specification), regions or provinces (NUTS code) and municipalities (LAU code). This part must also contain the start and end date and time (in UTC) of the event and hazard classification according to the Integrated Research on Disaster Risk peril classification and hazard glossary (IRDR, 2014), including the family, the main event and the peril type.

• Fieldwork information. It describes specific data about team members, including their affiliation and email address. Moreover, date and time of the visits, estimation of the fieldwork coverage over the total affected area and a brief description of difficult access areas should also be provided.

• Initial sources of information. It contains information available (web pages and links) on media and social networks and developed funnel cloud images (if any), together with a brief explanation of the initial information gathered before starting the damage survey.

• Meteorological conditions. This part describes weather conditions before, during and after the event according to direct witnesses, the visibility (darkness, precipitation), AWS data (location and a summary of the most relevant recorded data) and other data of interest derived from an overview of remote-sensing tools.

• Damage observed. A general description of the observed damage is given (i.e. the most common and the most relevant seen during the fieldwork), including the maximum DoD for every DI noticed.

• Direct witness inquiries. This part summarizes witness enquiries (which should also be attached entirely apart). It should contain, if available, the duration of the strong winds and a brief description of the experience of each witness.

• Characterization of the event. This final section contains the length and average and maximum width of the damage swath, the maximum wind intensity (specifying the intensity scale used), the translational direction and other data of interest such as the convective cell translation velocity.

### 3.3.2 GEOLOCATED INFORMATION TABLE

A geolocated information table providing disaggregated data for each point-of-damage is proposed, similarly as in Holzer et al. (2018). It should contain all relevant geolocated information gathered during the fieldwork and also damage locations provided by local authorities, emergency services, media and social networks, which have been previously collected and analysed. To better organise the information displayed, seven different location types (L1 to L7, see Table 4) are considered. Note that L1 to L3 (vegetation and man-made structures) correspond to points-of-damage locations so that, if possible, they should include information about intensity rating (DI-DoD), according to Sect. 3.2. The rest of locations describe positions of AWS, witnesses, pictures or wind-borne debris.

### 3.3.3 DATA LOCATION MAP

The third deliverable consists of a map or a KML file format containing geolocated information gathered during the field survey in order to allow further graphical analysis, for example using Google Earth software (Gorelick et al., 2017). It is proposed that each of the seven location types presented in Sect. 3.3.2 are represented with a different icon, with a specific colour for points-of-damage (L1 to L3 from Table 4) depending on its intensity. Moreover, in case of damage in trees with fall direction (L1) it is convenient to display on the map an arrow icon, whose direction should be the fall direction. Thereby, a damage tree

pattern analysis to discriminate between damage caused by a tornado or by a downburst should be easily carried out. Damage swath characteristics (length and width) should also be calculated using the data location map.

As an example, Fig. 7 shows the data location map of part of the fieldwork carried out on 25 March 2012 to study the EF1 tornado that affected the municipalities of Castellnou de Seana and Ivars d'Urgell (Catalonia) on 21 March 2012 (Bech et al., 2015). It displays the information contained in a fallen tree damage-point type (in this case, latitude, longitude, tree

fall direction, DI-DoD, a brief description and a photo) and in the non-official Ivars d'Urgell AWS location (here, latitude, longitude, AWS type and maximum wind speed plot), which registered a maximum wind gust of 26.4 m s$^{-1}$ during the event.

### 4 DISCUSSION

The proposed methodology is formulated in a convenient, feasible and detailed way so it can be readily used, but its practical application may present some weaknesses. Among the advantages of the proposed methodology are the relative simplicity of

340 the devices required (Table 1), which are not neither unusual nor expensive tools, so meteorological services, public research institutions and private entities may perform systematically damage surveys of reported events, analysing even suspicious developed funnel clouds from which it is previously unknown if reached the ground. Moreover, the easy to reproduce fieldwork process, and the generation of the proposed three final deliverables support the main objectives of the in-situ damage assessment, which are identifying the phenomenon type, estimating wind intensity and characterising the event and the damage swath.

Besides, the methodology is also intended to optimize time during in-situ measurements in order to make possible visiting the whole affected area as soon as possible to avoid the alteration of the damage scenario by clearing services, as commented in Sect. 3.1.

Surface in-situ analysis provides more detailed information than aerial surveys or analysis based only in remote sensing data. For instance, minor damage on vegetation and on man-made structures are more easily detected (e.g., Marshall et al.,

2012). In addition, it is possible to study in detail the soil state in forest areas and the degraded state or previous weaknesses of damaged elements, which are essential to assess the wind intensity, and measuring data of interest such as snapped trunk diameter or small wind-borne debris weight. On the other hand, tornado outbreaks and widespread events (such as derechos – e.g., Peterson, 2019; Chmielewski et al., 2020) may be cases where it is challenging to apply the methodology. Nevertheless, in Sect. 3.2 some clues to mitigate these problems have been provided, such as making a discontinuous analysis in forest areas

studying transversal stripes along the damage track every 250 to 500 metres if possible. Another option would be to distribute areas to be analysed among the members of the surveyors team to carry out several fieldworks in parallel. Especially in those

cases, and also in complex terrain events, aerial imagery could be useful to complement surface data, providing an overview of the damaged area and information from difficult access zones.

On the other hand, the geolocation of damaged elements and data of interest has a strong dependence on GPS signal reception. Geolocation accuracy depends on a number of factors including local terrain geometry, quality of the receiver antenna system or number of satellites observed. Photo cameras and smartphones have location errors usually ranging from 5 to 20 meters, typically being greatest in deep valleys, or close to large buildings or structures blocking satellite signals. To minimize geolocation errors, it is recommended to check the accuracy with manually selected reference locations and, if necessary, to correct damage locations on the summary map and on the geolocated information table. This is feasible in urban or periurban areas, where buildings or other elements are easily identifiable using high-resolution aerial images such as orthophotos, but not in forests or other natural areas without evident references where this verification may not be possible.

The estimation of wind intensity of convective origin is based on damage-rating scales, which relate the damage observed with the wind speed. Despite the proposed methodology is illustrated using the EF-scale, it should be noted that other intensity scales could be used such as the TORRO scale (Meaden et al., 2007). The practical application of the EF-scale has some limitations (Doswell et al., 2009), in spite of the progress made some years ago by introducing a more detailed intensity-rating scale (WSEC, 2006) compared to the original and simpler Fujita scale (Fujita, 1981, 1992). The Enhanced Fujita scale, developed in the USA, is mainly based on the damage caused by wind on standard US buildings and elements (schools, hospitals, automobile showrooms, etc.), so-called Damage Indicators (DI). When applied to areas outside the USA many DI may not exist, hampering its application as discussed in detail in Feuerstein et al. (2011) and Holzer et al. (2018). Moreover, there are elements which are susceptible to be damaged such as traffic signals, walls and fences, trash bins and vehicles, which are not included on the EF-scale.

The data gathered can have several uses, apart from contributing to build up homogeneous severe weather databases, which at the same time enhance the knowledge about tornado, downburst and straight-line winds occurrence. Insurance and reinsurance companies can be one of the major benefitted sectors from results of fieldworks, which usually need to know the affected area by a strong-convective wind event and its intensity to cover compensations and, in some specific cases as in Spain, the phenomenon type (De Groeve et al., 2014).

Data collected from damage on buildings (general photos and detailed pictures of deficiencies or previous weaknesses) may also contribute to study exposure and vulnerability of constructions in an area of interest and also to assess the failure modes (e.g., north-eastern Italy; Zanini et al., 2017; Pipinato, 2018), as it is pointed out in De Groeve et al. (2013, 2014). Moreover, the identification of typical damaged buildings using the information provided in the set of final deliverables can give a guideline for adapting intensity-rating scales, such as the EF-scale, out of the USA, solving partially those deficiencies on assessing wind intensity. In this work line, some authors have propounded new Damage Indicators to append to the above-mentioned scale using information derived from tens of fieldworks (Mahieu and Wesolek, 2016), to adapt them to typical man-made structures from other countries, as recently reported in Canada (Environment Canada, 2013) or Japan (Japan Meteorological Agency, 2015), or even to develop a standardised International Fujita Scale, as proposed in Groenemeijer et al. (2019). Furthermore, in some articles it has been discussed how to assess wind intensity throughout effects in vehicles, with data given by fieldworks

(Paulikas et al., 2016), similarly than presented here. As provided in the geolocated information map, where each point-of-damage with its DI-DoD pair is given, it is possible to assess the degree of damage severity along the damage swath of a tornadic event. This information can be very valuable to analyse the impact of tornadoes in future projected scenarios, for example modelling intensity damage areas in tornado paths using the data provided by fieldworks and assessing the possible consequences under different expected urban conditions (Ashley et al., 2014; Rosencrants and Ashley, 2015).

## 5 Summary and concluding remarks

In-situ damage survey data are used to study the consequences of natural hazards, such as floods or strong-convective damaging winds. The latter can be specifically characterized carrying out fieldworks, estimating the damage path length and width, and also the intensity of the event. Moreover, throughout an analysis of the data gathered it might be possible to clarify which phenomenon caused the damage (tornado, downburst or straight-line winds) in case neither images nor direct witness reports exist.

The purpose of this article is to provide an easily reproducible methodology to carry out surface strong-convective wind event damage surveys, which optimize time and economic resources. It is mainly based on collecting geolocated information about damaged man-made structures and vegetation, with the final aim of representing the damage scenario to study the event from a meteorological point of view. Complementary data from AWS close to the affected area and witness reports should also be gathered if available, and remote-sensing data should be used to get a deeper understanding of the convective storm event. With all this information, three final deliverables are generated (a standardised text report of the event, a table consisting of detailed geolocated information, and a map or a file in KML format). The whole data set allows further analysis and achieve purposes.

This methodology is based on previous studies and has been refined during the elaboration of 136 strong-convective wind damage surveys carried out in Spain between 2004 and 2018. Known limitations of its application include geolocation errors of damage, applicability of the EF-scale outside the USA and difficulties on analysing extensive events or complex topography areas. Nevertheless, surface-based detailed data provided, such as previously degraded state of damaged elements, minor damage on man-made structures and vegetation, snapped trees trunk diameter and soil state in forest areas, might be helpful to better analyse event consequences compared to other methodologies. In any case, the field survey data obtained are valuable for further analysis, complementing meteorological detailed case studies based on operational remote sensing such as Doppler weather radar data, surface observations and Numerical Weather Prediction model output. Moreover, the methodology proposed may contribute to standardise detailed field surveys, which are essential to build up and maintain robust and homogeneous databases of severe weather phenomena.

*Data availability.* The data used in this paper is available from the authors upon request.

## Appendix A: Tornado vs. downburst damage patterns

The determination of the damaging wind phenomenon (tornado, downburst or straight-line winds) can be rather challenging in some cases. As reported in previous studies (Hall and Brewer, 1959; Holland et al., 2006; Bech et al., 2009; Beck and Dotzek, 2010; Rhee and Lombardo, 2018), it can be assumed that the direction of fallen trees indicate the direction of maximum wind speed in strong-convective wind events, provided there are no influences from the terrain (i.e. slope favouring a specific fall direction) or from another tree fall interacting with the tree considered. Despite real wind damage patterns can be very complex due to its interaction with topography or with other nearby events (Forbes and Wakimoto, 1983; Cannon et al., 2016), theoretical-idealized damage swath patterns of both tornado and downburst wind fields can be compared with observed damage patterns in order to look for similarities to assess their possible origin.

As explained in previous studies (e.g., Holland et al., 2006; Bech et al., 2009), a simple approximation to describe a tornado vortex wind field near the surface is given by the Rankine vortex model. This approach combines an inner rigidly rotating core with an outer region with decreasing rotation speed. The wind field velocity module is defined in polar coordinates by Eq. (A1):

$$v(r) = \begin{cases} \frac{v_{max}r}{R} & \text{if } r \leq R \\ \frac{v_{max}R}{r} & \text{if } r > R \end{cases} \tag{A1}$$

where $v(r)$ is the wind velocity in function of the distance to the centre of the vortex $r$, $v_{max}$ is the maximum wind velocity, and $R$ is the vortex radius where $v(r) = v_{max}$.

Note that according to Eq. (A1), the Rankine vortex can describe in simple terms only a rotating vortex and its nearby environment, i.e. a stationary vortex. To model real tornadoes, a Rankine vortex with both tangential and radial wind components is combined with a translational movement, i.e. a homogeneous wind field. As described in Bech et al. (2009), according to Peterson (1992), two parameters are used to characterize this model: parameter $G$, which is the ratio between tangential velocity and translational velocity, and parameter $\alpha$, which is the angle between radial velocity and tangential velocity, corresponding 0° to a pure inflow, 90° to a pure tangential case and 180° to a pure outflow.

Examples of two-dimensional wind fields with different parameter configurations are shown in Fig. A1, including also their associated damage swath pattern shown as a rectangular panel below each two-dimensional wind field. The damage swath pattern is obtained computing the maximum wind vector of the wind field along the y axis, as the examples assume a northern translation of the vortex. In the first row (Fig. A1a, b, and c panels), translational velocity is 1/4 tangential velocity ($G = 4$) and, in the second row (Fig. A1d, e and f panels), translational velocity is equal to tangential velocity ($G = 1$).

In Fig. A1a, where tangential and inflow velocities are equal ($\alpha = 45°$), a convergence damage pattern is identified, whereas in Fig. A1b, where the radial component is zero ($\alpha = 90°$, i.e. pure tangential flow), the damage swath presents a rotational pattern. Fig. A1c presents pure outflow with no tangential velocity ($\alpha = 180°$), exhibiting a similar divergence pattern as Fig. A1f, in the damage swath, which could correspond with a classical downburst pattern.

Thus, based on this simple model, if fallen trees patterns present convergence or rotation, it can be assumed that a vortex caused the damage, whereas a divergent pattern would suggest the effects of a downburst. Similarly, the way debris is spread or

how a roof is collapsed or lifted can indicate winds with either a rotation and upward pattern (i.e. a tornado), or with a divergent and downward pattern (i.e. a downburst) – see Rhee and Lombardo (2018) for a more detailed discussion.

Nevertheless, it is also noticeable that in cases where tangential and translational velocities are similar ($G \approx 1$, see for example the second row of the Fig. A1), damage swaths may present only little differences among them. This can occur in weak (EF0 or EF1) tornado or downburst events that affect a small area. In these cases, damage also may be sparse, scattered and unconnected, which makes unidentifiable any damage pattern consistent with a tornado or a microburst (Bech et al., 2009; Rhee and Lombardo, 2018). Then, even a detailed damage survey, if there are neither images nor direct witnesses, it may not be sufficient to determine which type of phenomena caused the damage. This situation of inconclusive results regarding the phenomenon type occurred in 7 % of the 136 damage surveys carried out in Spain by the authors between 2004 and 2018.

As a real example, the case shown in Fig. A2 presents fallen poplar trees following a convergence pattern: in the right-half side from the damage swath, trees are blown down to the west, whereas in the left-half side they are uprooted to the north. Comparing this real case and idealised cases (Fig. A1), this damage pattern matches well the damage swath caused by a vortex with $G = 4$ and $\alpha = 45°$ (Fig. A1a). This fact along with other evidences confirm the hypothesis that damage was caused by a tornado, as presented in the Supplementary material. Moreover, it is remarkable that these vortex characteristics are also coherent with the damage rated as the lower EF1 bound and the mean translational velocity of 12 m s$^{-1}$, estimated using radar data from the Meteorological Service of Catalonia (not shown).

*Author contributions.* All authors contributed equally on this work.

*Competing interests.* The authors declare that they have no conflict of interest.

*Acknowledgements.* The authors gratefully acknowledge individuals supporting wind damage assessments carried out during these years, especially to Joan Arús, Andrés Cotorruelo, Petra Ramos and particularly to Miquel Gayà for his pioneer systematic studies of tornadoes in Spain, the Consorcio de Compensación de Seguros (CCS) support, and also three anonymous reviewers for their comments and suggestions that improved the present article. This research was performed with partial funding from projects CGL2015-65627-C3-2-R (MINECO/FEDER), CGL2016-81828-REDT (AEI) and RTI2018-098693-B643-C32 (AEI), and also from the Water Research Institute (IdRA) of the University of Barcelona.Antonescu et al. (2017)

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

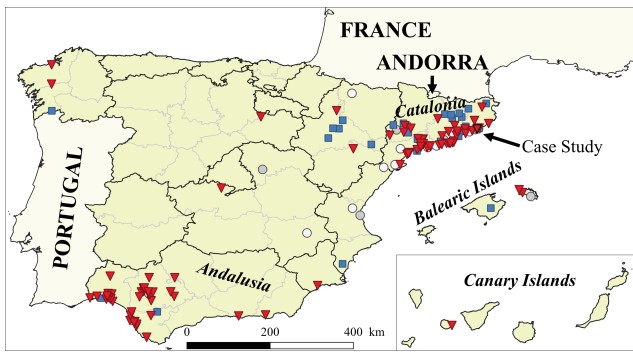

**Figure 1.** Location of 136 analysed events in Spain using the proposed methodology between 2004 and 2018, mostly concentrated in Andalusia and Catalonia. Symbols indicate locations of tornadoes (red triangles), downbursts (blue squares), undetermined phenomena (grey circles) and other phenomena such as gust fronts, funnel clouds which did not touch down, or dust devils (white circles). The case study location for which final deliverables are attached as Supplementary material is indicated on the map. Black contours delimitate regions and grey lines, provinces.

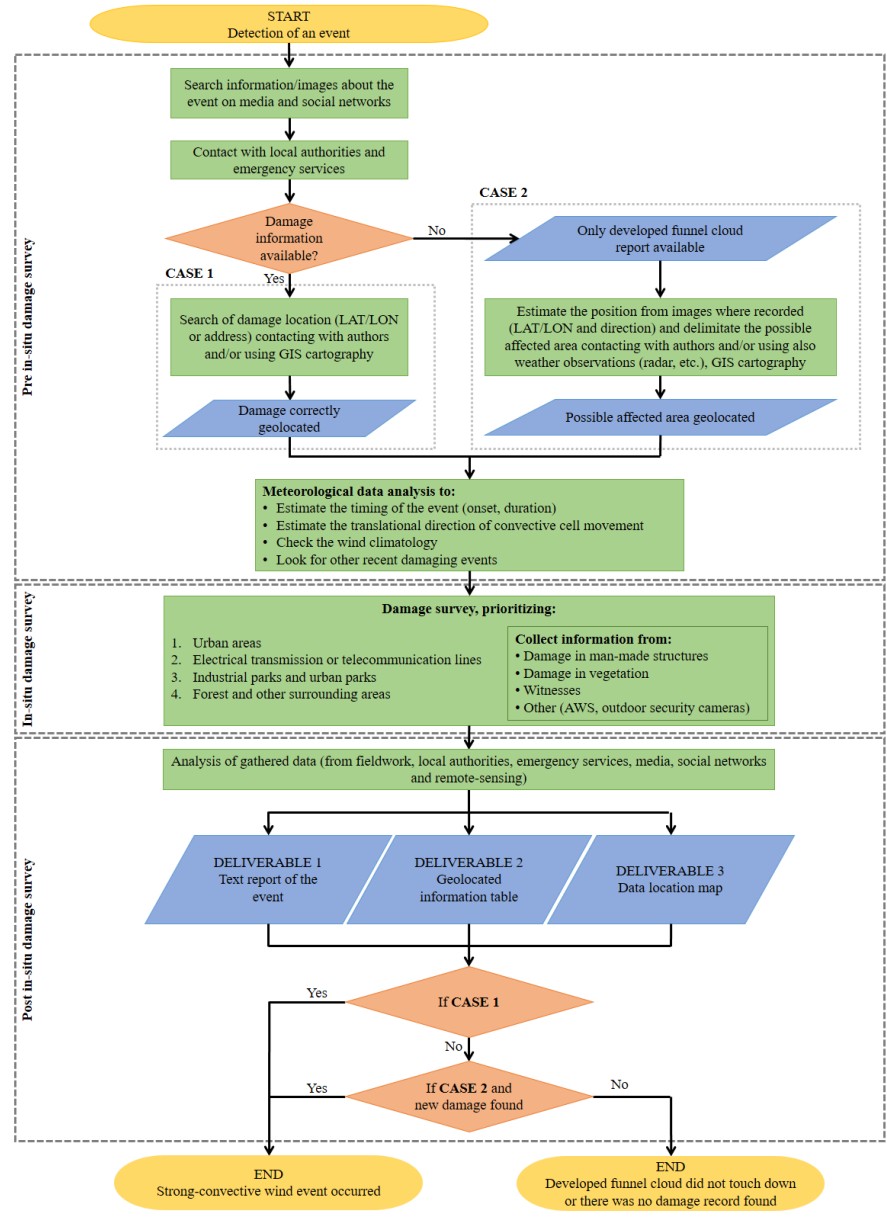

**Figure 2.** Flow diagram of the structure and application of the proposed methodology to carry out strong-convective winds fieldwork damage assessment. Start and end are shaded in yellow, processes in green, decisions in orange and inputs/outputs in blue.

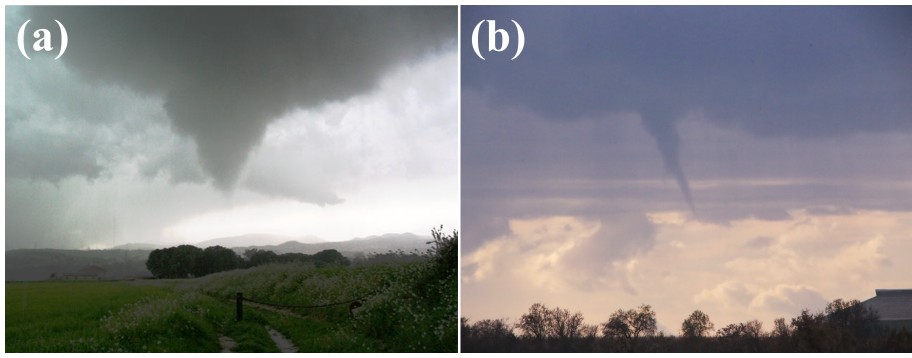

**Figure 3.** (a) Well-developed funnel cloud observed in Santa Eulàlia de Ronçana (Catalonia) on 4 April 2010 (Author: @CalabobosChaser), and (b) Well-developed funnel cloud observed in Bellpuig (Catalonia) on 1 December 2017 (Author: Edgar Aldana). In both cases no evident tornado was actually observed (i.e. touchdown) but nearby damage was reported suggesting tornado occurrence.

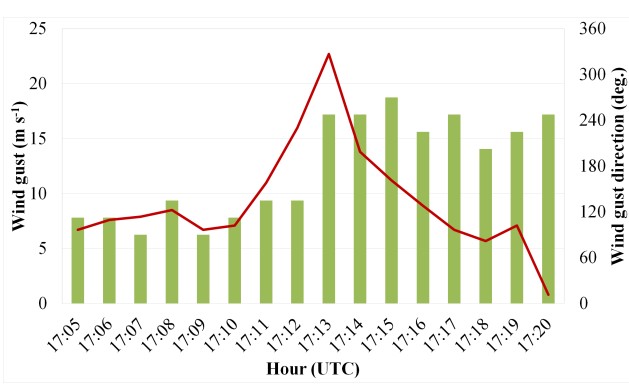

**Figure 4.** Wind gust (red line) and wind gust direction (green bars) registered by an AWS in Mataró (Catalonia) with 1 minute temporal resolution data. The AWS was located 240 m west of the estimated centre of the EF0 tornado track, on 23 November 2016. Data source: Meteomar, Consell Comarcal del Maresme.

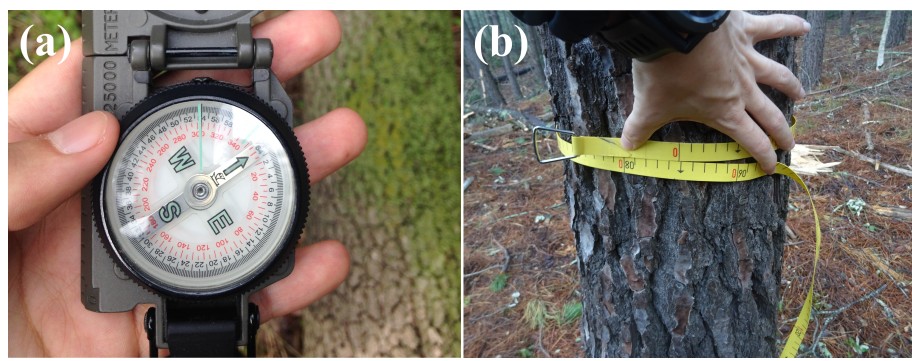

**Figure 5.** (a) Measure of the fall direction of a tree and (b) measure of trunk diameter during the damage survey of an EF1 tornado on 13 October 2016 in Llinars del Vallès (Catalonia) and an EF2 tornado on 7 January 2018 in Darnius (Catalonia), respectively (Author: Oriol Rodríguez).

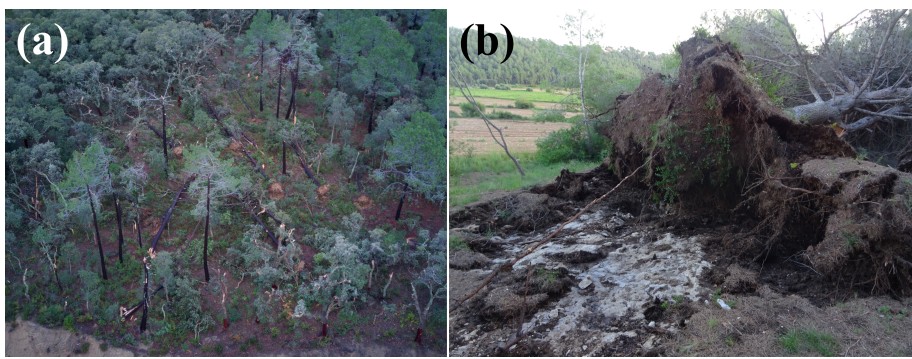

**Figure 6.** (a) Drone image of a mixed Mediterranean forest in Darnius (Catalonia) where most pine trees were blown down whereas cork oaks were only slightly affected with broken branches by an EF2 tornado, on 7 January 2018 (Author: Jonathan Carvajal). (b) Pine blown down by an EF0 tornado in Perafort (Catalonia) in a very thin, moist soil area, on 14 October 2018 (Author: Oriol Rodríguez).

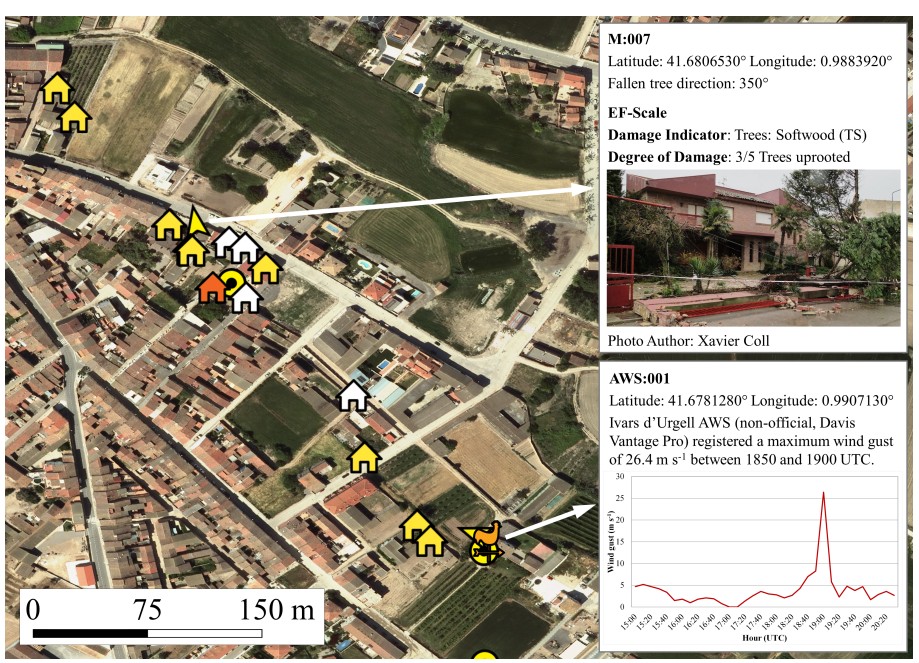

**Figure 7.** Data location map and two examples of recorded information from the 21 March 2012 EF1 Ivars d'Urgell (Catalonia) tornado track. Map symbols indicate locations of AWS (orange weathervane), damage in man-made structures (house icon) and fallen trees or damaged vegetation elements (arrow and circle icon if no direction is available, respectively). Icon colours indicate damage intensity using the EF-scale: EF0 (yellow), EF1 (orange) and unrated (white). The background orthophoto is from the Institut Cartogràfic i Geològic de Catalunya (ICGC), http://www.icc.cat (last access: September 2019), under a CC BY 4.0 license.

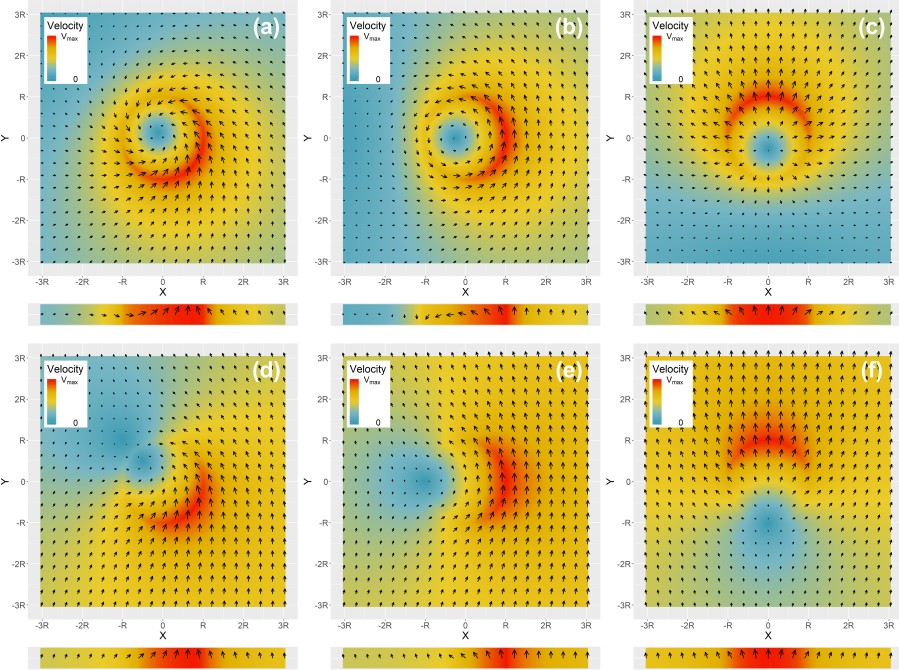

**Figure A1.** Two dimensional near surface horizontal wind fields and damage swaths for the cases: (a) $G = 4$ and $\alpha = 45°$, (b) $G = 4$ and $\alpha = 90°$, (c) $G = 4$ and $\alpha = 180°$, (d) $G = 1$ and $\alpha = 45°$, (e) $G = 1$ and $\alpha = 90°$, and (f) $G = 1$ and $\alpha = 180°$. Adapted from Figures 3 and 4 of Bech et al. (2009).

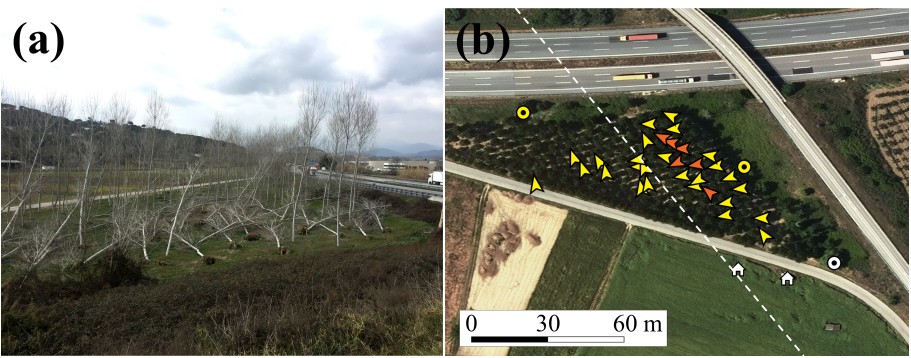

**Figure A2.** (a) A poplar plantation from Fogars de la Selva (Catalonia) affected by the 15 October 2018 EF1 Malgrat de Mar – Massanes tornado, and (b) fallen trees directions of the same poplar plantation. Map symbols indicate locations of damage in man-made structures (house icon) and fallen tree or damaged vegetation element (arrow or circle icon if no direction is available). Icon colours indicate damage intensity: EF0 (yellow), EF1 (orange) and unrated (white). The white discontinuous line separates the right-half and the left-half sides of the damage swath where predominant fall tree direction are west and north, respectively. The background orthophoto is from the Institut Cartogràfic i Geològic de Catalunya (ICGC), http://www.icc.cat (last access: September 2019), under a CC BY 4.0 license.

**Table 1.** Devices required to perform strong-convective wind damage surveys.

| Device reference | Device |
| --- | --- |
| D1 | Smartphone or camera with GPS image geolocation and orientation (azimuth pointing) capabilities |
| D2 | Compass |
| D3 | Tape measure |
| D4 | Hand-counter |
| D5 | Suitcase balance |

**Table 2.** Variables and maximum uncertainties recommended of data descriptors for damaged man-made structures and vegetation elements. The first four variables are required for all damaged elements (both man-made structures and vegetation). Dragged distance, direction and weight of wind-borne debris should be measured if possible for relevant and representative elements (e.g., fragment of panel roof). Fallen tree direction and trunk diameter should be measured in case of uprooted and snapped trees, respectively. Degraded state or previous weakness of damaged elements should also be reported.

| Index | Variable | Uncertainty | Comments |
|---|---|---|---|
| V1 | Latitude | $\pm\ 1\times10^{-4}$ deg. | Measured with GPS camera. |
| V2 | Longitude | $\pm\ 1\times10^{-4}$ deg. | Measured with GPS camera. |
| V3 | Damage Indicator (DI) | — | Determined during the post in-situ damage survey using intensity-rating scales as EF-scale. |
| V4 | Degree of Damage (DoD) | — | Determined during the post in-situ damage survey using intensity-rating scales as EF-scale. |
| V5 | Fallen tree direction | $\pm\ 5$ deg. | In case of uprooted trees. Measured with a compass. |
| V6 | Dragged direction object | $\pm\ 5$ deg. | Direction of the displacement. Measured with a compass or GIS tools. |
| V7 | Trunk diameter | $\pm\ 5$ cm | In case of snapped trees. The trunk perimeter is measured with a tape measure and then the diameter can be calculated. |
| V8 | Dragged distance object | $\pm\ 1$ m | Distance between the final position and the origin of an object displaced by the wind. Measured with a tape measure or GIS tools. |
| V9 | Weight of wind-borne debris | $\pm\ 10$ % | Weight of an object of interest moved by the wind. In case of small objects, measured with a balance if possible. |
| V10 | Previous weakness | — | Description of deficiencies that can increase the vulnerability of elements to strong winds. |

**Table 3.** Witness questionnaire (reference and question).

| Question reference | Question |
| --- | --- |
| Q1 | At what time did the phenomenon occur? |
| Q2 | Where were you when the phenomenon took place? |
| Q3 | How long did the strongest winds last? (Some seconds, around one minute, several minutes...). |
| Q4 | During the phenomenon, did you hear any special or rare noise? |
| Q5 | How was the weather like before, during and after the phenomenon? (Light rain, heavy rain, small hail, large hail, snow, no precipitation). |
| Q6 | Have you noticed other areas with damage? |
| Q7 | Do you remember any similar phenomenon in this area before? |

**Table 4.** Information location types (reference, description and data that should be presented).

| Location reference | Description | Data |
|---|---|---|
| L1 | Damage in trees with fall direction | Latitude, longitude, DI-DoD, previous weaknesses, fall direction |
| L2 | Damage in trees without fall direction | Latitude, longitude, DI-DoD, previous weaknesses, trunk diameter (if snapped tree) |
| L3 | Damage in man-made structures | Latitude, longitude, DI-DoD, previous weaknesses |
| L4 | AWS location | Latitude, longitude, data (maximum wind gust, direction of maximum wind gust and hour) |
| L5 | Witness location | Latitude and longitude of the witness location at the moment of the meteorological event and a brief description of his experience |
| L6 | Image of the phenomenon | Latitude and longitude of the point where image was recorded and orientation |
| L7 | Wind-borne debris | Latitude, longitude, distance and direction of the displacement, size and weight of the object if measured |