# Peer review of "A methodology to conduct wind damage field surveys for high impact weather events of convective origin"

_Natural Hazards and Earth System Sciences, 2019_

## Referee Comment (RC1) · Anonymous Referee #1 · 15 Oct 2019

I really admire the scientific interest and the passion of authors on severe storms and on understanding their associated damages. Unfortunately, the submitted manuscript, describing a methodology to conduct wind damage surveys, is not ready for publication. My opinion is based on the following comments: 1) the manuscript does not fit within the scope of the journal; 2) the aim of the manuscript is not clearly stated: why is this methodology necessary? what are the new ideas that the authors are bringing on conducting wind damage surveys? how is this methodology relevant for other regions of Europe besides Spain? 3) some of the recommendations for wind damage survey are common sense (but I do understand that is necessary to have a document collecting all of them), but most of them are just vague and not clearly explained; how are

the forest damage assessments used to differentiate between damages produces by tornadoes and other types of convective winds? 4) there is no clear mentioning in the entire manuscript on how to use the proposed methodology to differentiate between damages associated with different types of convective winds events; 5) the manuscript is not very well structured and the English is not of publication quality and requires major improvements.

I do encourage the authors to further develop, refine, and describe their methodology in more details and to make it available as a report to the scientific community.

---

## Referee Comment (RC2) · Anonymous Referee #2 · 22 Oct 2019

The submitted manuscript is based on very important work, but the manuscript itself is not ready for publication.

I thank the authors for taking this step, and I value their wish to contribute to a proper standardization of working methods.

My suggestion is to write a more focused and condensed paper on the methodology. Details on practical approaches and case examples could be added separately as an attachment file.

Literature search shows that in this journal only last year important aspects of this topic were discussed in the following paper, namely "(2) to propose a repeatable working method for assessing damage and reconstructing the path and magnitude of local windstorm and tornado cases": 04 Jun 2018 A forensic re-analysis of one of the deadliest European tornadoes Alois M. Holzer, Thomas M. E. Schreiner, and Tomáš Púčik Nat. Hazards Earth Syst. Sci., 18, 1555–1565, https://doi.org/10.5194/nhess-18-1555-2018, 2018

In the submitted manuscript there is no reference to this work or no discussion about the previously proposed working method in comparison to the newly proposed one. Such comparison or discussion could help make a newly submitted version useful for the severe storms community, and I therefore encourage the authors to submit such a more streamlined paper.
* * *

---

## Referee Comment (RC3) · Anonymous Referee #3 · 25 Oct 2019

As authors state in the introduction, the objective of the paper is "to propose a methodology to conduct wind-field damage assessments of convective-driven events in a systematic way, to contribute to the creation and maintenance of homogeneous databases". Accordingly, the authors present first the methodology they propose, followed by its critical analysis and by two real implementations.

The objective of the paper, as stated in the introduction (note that in the conclusions the objective changes in "to provide guidelines for gathering pictures and locations of damage on manmade structures and on vegetation, using smartphones or photo cameras with geolocation capability"), is in the scope of the journal and is also related

to a very important weakness of natural hazards research and practice, being the lack of standardised data on past hazardous events. Still, the paper suffers from many criticalities, which prevent its publication in the present form. In the following, such criticalities are explained in detail while I did not supply specific comments, at this stage of the review.

Main criticalities

Methodology. I really appreciate all the anecdotal experience put in the paper by authors. Still, the methodology lacks of a clear logical structure; for example, some of the tasks included in the section 2.1 "survey planning" (e.g. gathering information and pictures on damage location on the media and social networks) are also included in the preliminary considerations discussed in the section methodology. Likewise, the section called "previous considerations" seems more related to preliminary considerations. With respect to this point, I think that a flowchart of the methodology, showing its steps in a logical order could support both possible users and readers. Moreover, the proposed methodology is not a systematic or a standardised one: only some indications of which could be the different aspects to be surveyed are provided (see e.g. section 2.2.2) without a systematic and standardised procedure for their survey and collection (e.g. by means of pre-defined questions in a form). The only "step" that, in some way, is standardised is the witness enquiries (section 2.2.4), for which pre-defined questions are provided. The lack of standardisation is a big limit towards the objective of creating homogenous databases, given that the parameters/aspects to be surveyed, the way they must be surveyed/measured, and the possible values assumed by each of them is a subjective choice of the surveyor.

Objective of the survey. The objective of the survey is not really clear. Is it reproducing the damage scenario? Is it identifying the kind of event for insurance purposes? Given the effort requires by on-field surveys, I think that the multi-usability of collected data should be pursued (see references below). For example, what about the amount of damage data collected? Are they used only to characterise the hazard? This arises

also the question of whether the products (deliverables) identified in the paper are suitable for multi-purposes uses of data

Case studies. Case studies do not supply examples of how to implement the methodology but simply describe the events and the scenarios resulting from the survey. I think this is due to the lack of standardised tools for the implementation of the methodology previously commented.

References

De Groeve T., Poljansek K. & Ehrlich D. Recording disasters losses: recommendation for a European approach. JRC Scientific and Policy Report [online]. 2013. Available at: http:// publications.jrc.ec.europa.eu/repository/bitstream/ 111111111/29296/1/lbna26111enn.pdf.

De Groeve T., Poljansek K., Ehrlich D. & Corbane C. Current status and best practices for disaster loss data recording in EU Member States. JRC Scientific and Policy Report [online]. 2014. Available at: http://publications.jrc.ec.europa.eu/ repository/bitstream/JRC92290/lbna26879enn.pdf

---

## Author Comment (AC3) · 13 Dec 2019

**Reply to Anonymous Referee #3**

We thank Anonymous Referee #3 for reviewing our manuscript "A methodology to conduct wind damage field surveys for high impact weather events from convective origin". We believe that the comments provided help to reconsider the structure of the text and improve the clarity of some aspects of the manuscript, particularly to explicitly mention that our study is focused on meteorological aspects which may complement other general features of general post-disaster damage surveys. We provide an item-by-item reply below:

As authors state in the introduction, the objective of the paper is "to propose a methodology to conduct wind-field damage assessments of convective-driven events in a systematic way, to contribute to the creation and maintenance of homogeneous databases". Accordingly, the authors present first the methodology they propose, followed by its critical analysis and by two real implementations.

(1) The objective of the paper, as stated in the introduction (note that in the conclusions the objective changes in "to provide guidelines for gathering pictures and locations of damage on manmade structures and on vegetation, using smartphones or photo cameras with geolocation capability"), is in the scope of the journal and is also related to a very important weakness of natural hazards research and practice, being the lack of standardised data on past hazardous events. Still, the paper suffers from many criticalities, which prevent its publication in the present form. In the following, such criticalities are explained in detail while I did not supply specific comments, at this stage of the review.

Reply: thanks for this comment. As it is explained, the main objective of the paper presented on the introduction is "to propose a methodology to conduct wind-field damage assessments of convective-driven events in a systematic way, to contribute to the creation and maintenance of homogeneous databases". We will rewrite this on conclusions to "provide a systematic and easily-reproducible methodology to carry out strong-convective wind event damage surveys, mainly based on gathering geolocated information about damaged man-made structures and vegetation, with the final aim of representing the damage scenario to study the event from a meteorological point of view".

Main criticalities

(2) Methodology. I really appreciate all the anecdotal experience put in the paper by authors. Still, the methodology lacks of a clear logical structure; for example, some of the tasks included in the section 2.1 "survey planning" (e.g. gathering information and pictures on damage location on the media and social networks) are also included in the preliminary considerations discussed in the section methodology.

Reply: according to this and the next comment, and also to Referee #1, we will improve the structure merging "Survey planning" and "previous considerations" sections into

a new one titled "preliminary considerations", to strength the chronological order of the tasks to carry out to apply the proposed methodology.

(3) Likewise, the section called "previous considerations" seems more related to preliminary considerations. With respect to this point, I think that a flowchart of the methodology, showing its steps in a logical order could support both possible users and readers.

Reply: thanks for the recommendation respect to adding a flowchart. We propose to present it in Section 2 (Methodology), as an overview of the proposed methodology. Following, we attached the flowchart proposal.

[Figure]

Figure 1. Flow diagram of the structure and application of the proposed methodology to carry out strong-convective wind damage assessment.

(4) Moreover, the proposed methodology is not a systematic or a standardised one: only some indications of which could be the different aspects to be surveyed are provided (see e.g. section 2.2.2) without a systematic and standardised procedure for their survey and collection (e.g. by means of pre-defined questions in a form). The only "step" that, in some way, is standardised is the witness enquiries (section 2.2.4), for which predefined questions are provided. The lack of standardisation is a big limit towards the objective of creating homogenous databases, given that the parameters/aspects to be surveyed, the way they must be surveyed/measured, and the possible values assumed by each of them is a subjective choice of the surveyor.

Reply: Thanks for your comment. We agree that the methodology can be better explained. To clarify and standardize the methodology, we propose to list explicitly which data is required for every damage location. For man-made structures are:

- Latitude
- Longitude
- Damage Indicator – DI *(omit it in case of damaged element not contained in damage rating scales)*
- Degree of Damage – DoD *(omit it in case of damaged element not contained in damage rating scales)*
- Dragged distance and direction (in case of object displaced with a known origin) – *distance between the origin and the final position of the object measured with a measuring tape and direction of the movement measured from the initial position of the object with a compass (minimum resolution of 5 degrees)*
- Previous weakness (lack of anchors, weak structure, oxide)

And for damaged vegetation:

- Latitude
- Longitude
- Damage Indicator - DI
- Degree of Damage - DoD
- Fall direction (in case of uprooted tree) – *measured with a compass (minimum resolution of 5 degrees)*
- Trunk diameter (in case of snapped tree) – *perimeter measured with a measuring tape*
- Previous weakness (moist soil, rocky subsoil, lack of extended roots, old tree)

According to De Groeve *et al.* (2014) we will propose some changes in final deliverables to standardize them including elements from Table 2 of the above mentioned study, showed in pages 34 and 35.

We will also attach the final deliverables of a case study to show explicitly how collected data is organized and presented.

(5) Objective of the survey. The objective of the survey is not really clear. Is it reproducing the damage scenario? Is it identifying the kind of event for insurance purposes?

Reply: the objective of the survey is reproducing the damage scenario to study the event from a meteorological point of view. With the gathered data: (1) the type of strong-convective wind phenomenon is identified, (2) the damage swath is characterized (it is determined the length and the width of the track) and (3) the intensity of the phenomenon is also estimated. In the particular case of Spain, as it is mentioned in the paper, the reinsurance public company also needs to know which kind of phenomenon caused damage, but the main interest of carrying out field works characterize events to build up a robust database about this kind of natural hazard. We will add this information in the introduction to clarify it.

(6) Given the effort requires by on-field surveys, I think that the multi-usability of collected data should be pursued (see references below). For example, what about the amount of damage data collected? Are they used only to characterise the hazard? This arises also the question of whether the products (deliverables) identified in the paper are suitable for multi-purposes uses of data Case studies.

Reply: all the collected data is used to characterise strong-convective winds (i.e. phenomenon type, intensity, damage path length and width) with the main aim of building up a robust and homogeneous database of this kind of meteorological phenomena. Moreover, as it is mentioned in De Groeve *et al.* (2013) and in De Groeve *et al.* (2014), data gathered in a field work is also useful to further analyse the exposition and vulnerability of damaged man-made structures, and it is also possible to study the impact of strong convective wind phenomena in an area. In addition, all this information can also be used to enhance or compliment wind intensity rating scales (as presented in Mahieu and Wesolek, 2016), even to create a new one as it is proposed in Groenemeijer *et al.* (2018). These comments will be included in the corrected manuscript.

(7) Case studies do not supply examples of how to implement the methodology but simply describe the events and the scenarios resulting from the survey. I think this is due to the lack of standardised tools for the implementation of the methodology previously commented.

Reply: the main objective of analysing these two case studies was to explain in practical cases which handicaps we found and to show the final deliverables, similarly as it is done usually in meteorological case studies. Attending to Referee #2, we will attach the three deliverables from a case study as supplementary material. Thus, it will be shown explicitly how all the data gathered during a strong-convective wind damage survey carried out following the proposed methodology is organized and presented in a practical way.

References

De Groeve T., Poljansek K. & Ehrlich D. Recording disasters losses: recommendation for a European approach. JRC Scientific and Policy Report [online]. 2013. Available at: http://publications.jrc.ec.europa.eu/repository/bitstream/111111111/29296/1/lbna261 11enn.pdf.

De Groeve T., Poljansek K., Ehrlich D. & Corbane C. Current status and best practices for disaster loss data recording in EU Member States. JRC Scientific and Policy Report [online]. 2014. Available at: http://publications.jrc.ec.europa.eu/repository/bitstream/JRC92290/lbna26879enn. pdf

---

## Author Response (AR1)

**Reply to Anonymous Referee #1**

We thank Anonymous Referee #1 for reviewing our manuscript "A methodology to conduct wind damage field surveys for high impact weather events from convective origin". We believe that the comments provided identified points which were not clear enough and also helped us to reconsider the structure of the text. We provide an item-by-item reply below:

(1) the manuscript does not fit within the scope of the journal;

Reply: NHESS "Aims and scope" section (https://www.natural-hazards-and-earth-system-sciences.net/about/aims_and_scope.html) lists five paragraphs describing the journal scope, being the 4th "the design, development, experimentation, and validation of new techniques, methods, and tools for the detection, mapping, monitoring, and modelling of natural hazards and their human, environmental, and societal consequences". The manuscript, as indicated by its title, describes a methodology to perform wind damage field surveys so we believe it fits well under the journal scope explained in the paragraph quoted as it is a contribution to improve the detection, mapping and characterization of wind damage. This is important as allows to better characterize specific meteorological phenomena, with the particularities associated with damage from convective storms. The comment of the reviewer indicates that a clearer link between the topic and the scope of the journal should be explicitly explained, an aspect which it has been incorporated in the corrected version (see lines 55 to 57).

(2) the aim of the manuscript is not clearly stated: why is this methodology necessary? what are the new ideas that the authors are bringing on conducting wind damage surveys? how is this methodology relevant for other regions of Europe besides Spain?

Reply: The aim of the manuscript was introduced in lines 45-46 of the first submitted version: "The objective of this paper is to propose a methodology to conduct wind-field damage assessments of convective-driven events in a systematic way, to contribute to the creation and maintenance of homogeneous databases.". The methodology includes three deliverables which are a damage survey summary, a geolocated information table and a data location map, all of them described in Section 2.3. The presented methodology may contribute to homogenize the way of collecting information for studying strong-convective winds phenomena. Building up meteorological databases of severe weather events that discriminate between tornadic, downbursts and other convective winds, is a task which requires in-situ damage assessment data and this research tries to contribute to this objective. In the specific case of Spain, the interest to know which type of severe weather convective wind phenomenon damaged an area is not only from the meteorological point of view, but also from the public reinsurance perspective, as it is reported in De Groeve et al. (2014). In the new submitted version, we have explained this aspect more clearly in Section 1 (Introduction).

(3) some of the recommendations for wind damage survey are common sense (but I do understand that is necessary to have a document collecting all of them), but most of them are just vague and not clearly explained; how are the forest damage assessments used to differentiate between damages produces by tornadoes and other types of convective winds? There is no clear mentioning in the entire manuscript on how to use the proposed methodology to differentiate between damages associated with different types of convective winds events;

Reply: Thanks for indicating this issue; we thought that giving proper references was enough to cover this point but we understand the need to expand it briefly. As reported in other studies cited in the text (Hall and Brewer, 1959; Holland et al., 2006; Bech et al., 2009; Beck et al., 2010; Rhee and Lombardo, 2018), it can be assumed that fallen direction of trees indicate the direction of maximum wind speed in a strong-convective winds event. Knowing how is the damage swath pattern of both theoretical tornado and downburst and comparing it to the damage pattern found during a damage survey assessment in a forest area, one may estimate which phenomenon took place.

As explained in Bech et al. (2009), a simple approximation to describe a tornado vortex near the surface is given by the Rankine vortex model, which is defined in polar coordinates, by:

$$
\begin{cases}
v(r) = \dfrac{v_{max}r}{R} & \text{when } r \leq R \\[2mm]
v(r) = \dfrac{v_{max}R}{r} & \text{when } r > R
\end{cases}
$$

where $v(r)$ is the velocity in function of the distance to the centre of the vortex $r$, $v_{max}$ is the maximum velocity, and $R$ is the radius where $v(r) = v_{max}$.

To model tornadoes, a Rankine vortex with tangential and radial wind components is combined with a translational movement. As it is explained in Bech et al. (2009), two parameters are used, based in Peterson (1992a), to characterize this model: parameter G, which is the ratio between $v_{tang}$ and $v_{trans}$, and parameter α which is the angle between $v_{rad}$ and $v_{tang}$, corresponding 0° to pure inflow, 90° to a pure tangential case and 180° to a pure outflow.

In the figure attached below it is shown the two-dimensional wind field associated to three different vortex configurations and their theoretical damage swath pattern (as a rectangular panel below each two-dimensional wind field), which are described by the maximum wind vectors perpendicular to the translational movement. In the first column, translational velocity is 1/4 maximum tangential velocity (G = 4) and, in the second, translational velocity is equal to maximum tangential velocity (G = 1).

In case (a), where tangential and inflow maximum velocities are equal (α = 45°), a convergence damage pattern is identified, whereas in case (c), where the radial component is zero (α = 90°, i.e. pure tangential flow), damage swath presents a rotational pattern. In case (e), which presents pure outflow with no tangential velocity (α = 180°, i.e. downburst), there is a clear divergence in the damage swath pattern. Thus, based on this model, if fallen trees pattern presents convergence or

rotation, it can be assumed that a vortex caused the damage, whereas if it is divergent, it might be a downburst.

Nevertheless, it is also noticeable that in cases where tangential and translational velocities are similar (G ≈ 1, see for example the second column of the Figure), damage swaths may present only little differences. This can occur in weak (EF0 or EF1) tornado or downburst events. Then, even with a detailed damage survey, if there is no direct witness or image of the meteorological phenomenon, it may not be possible to know which type of phenomena caused the damage.

[Figure]

Figure. Two dimensional near surface horizontal wind fields and damage swaths for the cases (a) G=4 and α=45°, (b) G=4 and α=90°, (c) G=4 and α=180°, (d) G=1 and α=45°, (e) G=1 and α=90°, and (f) G=1 and α=180°. Adapted from Figures 3 and 4 of Bech et al. (2009).

In the new submitted version of the manuscript, this explanation and the Figure are included in Section 3.3 (Discussion – Tornado vs. downburst damage patterns).

(4) the manuscript is not very well structured and the English is not of publication quality and requires major improvements.

Reply: From the previous comment we agree that the original structure, based on the chronological order of steps carried out during damage surveys, can be improved. Maintaining previous elements of the structure, we propose to change some subsection parts to explain better their meaning. In particular, we formulate a three-part methodology centred on the in situ damage survey tasks (ISDS): pre-ISDS, ISDS and post-ISDS.

In the original structure, it is commented firstly how to prepare the visit to the affected area. After that, survey tasks are explained (previous considerations, how to conduct the man-made structures damaged analysis, the collection of vegetation damage data and how to perform witness enquiries). Then, the deliverables are presented (the damage survey summary, the geolocated information table and the data location map) and, finally, there is a brief discussion where three challenges of applying this methodology are explained. Nevertheless, according to this comment and to Referee #3 we propose to modify the manuscript structure by merging "Survey planning" and "previous considerations" sections into a new one entitled "Pre in-situ survey tasks" and by removing the two case studies from the manuscript. Moreover, the deliverables of the 15 October 2018 Malgrat de Mar-Massanes tornado case study are now attached as supplementary material to clearly illustrate this concept with a real example. Therefore, the new manuscript structure is as follows:

1. Introduction
2. Methodology
    2.1. Pre in-situ survey tasks
    2.2. In-situ survey tasks
        2.2.1. Man-made structures damage assessment
        2.2.2. Forest damage assessment
        2.2.3. Witness enquiries
    2.3. Post in-situ survey tasks and deliverables
        2.3.1. Damage survey summary
        2.3.2. Geolocated information table
        2.3.3. Data location map
3. Discussion
    3.1. Geolocation accuracy
    3.2. Damage intensity rating
    3.3. Tornado vs. downburst damage patterns
4. Summary

In addition, as Referee #3 suggested, we have added a flow diagram (see below) in Section 2 (Methodology) to clarify the structure and application of the proposed methodology.

[Figure]

Figure. Flow diagram of the structure and application of the proposed methodology to carry out strong-convective winds fieldwork damage assessment.

In the new manuscript version English have been revised and corrected.

 I do encourage the authors to further develop, refine, and describe their methodology in more details and to make it available as a report to the scientific community.

Reply: Thank you very much for your comments and encouragement. With your -and the rest of reviewers- suggestions, we have worked on an improved version of the manuscript for a new submission.
We thank Anonymous Referee #2 for reviewing our manuscript "A methodology to conduct wind damage field surveys for high impact weather events from convective origin". We believe that the comments provided help to reconsider the structure of the text and improve the clarity of some aspects of the manuscript. We provide an item-by-item reply below:

The submitted manuscript is based on very important work, but the manuscript itself is not ready for publication. I thank the authors for taking this step, and I value their wish to contribute to a proper standardization of working methods.

(1) My suggestion is to write a more focused and condensed paper on the methodology. Details on practical approaches and case examples could be added separately as an attachment file.

Reply: Thank you for your constructive suggestion. According to this comment and Referee #3, in the new manuscript version we have provided, as an example, a set of deliverables (KML file, table and text summary) of the 15 October 2018 Malgrat de Mar-Massanes tornado case study as supplementary material, with the aim to better illustrate the results of the damage survey assessment. We have also been more focused on the methodology and the discussion, although we mention in the main text some practical approaches that we think that need to be taken into account.

(2) Literature search shows that in this journal only last year important aspects of this topic were discussed in the following paper, namely "(2) to propose a repeatable working method for assessing damage and reconstructing the path and magnitude of local windstorm and tornado cases": 04 Jun 2018 A forensic re-analysis of one of the deadliest European tornadoes Alois M. Holzer, Thomas M. E. Schreiner, and Tomáš Púcikˇ Nat. Hazards Earth Syst. Sci., 18, 1555–1565, https://doi.org/10.5194/nhess-18-1555-2018, 2018. In the submitted manuscript there is no reference to this work or no discussion about the previously proposed working method in comparison to the newly proposed one. Such comparison or discussion could help make a newly submitted version useful for the severe storms community, and I therefore encourage the authors to submit such a more streamlined paper.

Reply: thanks for considering this valuable reference. We agree completely on the interest of comparing both methodologies. Although in Holzer *et al.* (2018) a historical event is analysed, there are some similarities in how to proceed to collect and present data from recent events after carrying out an in-situ damage survey. In that paper, it is commented the importance of visiting the affected area as soon as possible to avoid the alteration of the scenario due to cleaning service tasks. They also propose to build up a database with DI and DoD for each damage object, whereas we propound to compliment it with in-situ measurements as weight, wind-borne distance of debris, and a description of previous weaknesses that could magnify damage. Moreover, in case of damaged trees, we also recommend to

measure trunk diameter (if snapped) to enhance the intensity rating, despite the difficulties of applying the EF-scale out of the United States. We also propose to record tree fallen direction (if uprooted), which is potentially useful to try to estimate which phenomenon affected the area in case of lack of direct witnesses or recorded images (this aspect is discussed in Section 3 in the new submitted version). We both mention the necessity to standardize tornado intensity throughout the world. The method for geo-reference photos of damage explained in Holzer *et al.* (2018) is similar to our proposal. In both cases, the objective is to estimate the location and direction where the author took the photo. In the new version of the paper we have developed these ideas comparing our proposal with that presented in the reference given.
We thank Anonymous Referee #3 for reviewing our manuscript "A methodology to conduct wind damage field surveys for high impact weather events from convective origin". We believe that the comments provided help to reconsider the structure of the text and improve the clarity of some aspects of the manuscript, particularly to explicitly mention that our study is focused on meteorological aspects which may complement other general features of general post-disaster damage surveys. We provide an item-by-item reply below:

As authors state in the introduction, the objective of the paper is "to propose a methodology to conduct wind-field damage assessments of convective-driven events in a systematic way, to contribute to the creation and maintenance of homogeneous databases". Accordingly, the authors present first the methodology they propose, followed by its critical analysis and by two real implementations.

(1) The objective of the paper, as stated in the introduction (note that in the conclusions the objective changes in "to provide guidelines for gathering pictures and locations of damage on manmade structures and on vegetation, using smartphones or photo cameras with geolocation capability"), is in the scope of the journal and is also related to a very important weakness of natural hazards research and practice, being the lack of standardised data on past hazardous events. Still, the paper suffers from many criticalities, which prevent its publication in the present form. In the following, such criticalities are explained in detail while I did not supply specific comments, at this stage of the review.

Reply: thanks for this comment. As it is explained, the main objective of the paper presented on the introduction is "to propose a methodology to conduct wind-field post-event damage surveys of convective-driven events systematically". We have rewritten this on conclusions to "provide a systematic and easily-reproducible methodology to carry out strong-convective wind event damage surveys, mainly based on gathering geolocated information about damaged man-made structures and vegetation, with the final aim of representing the damage scenario to study the event from a meteorological point of view".

Main criticalities

(2) Methodology. I really appreciate all the anecdotal experience put in the paper by authors. Still, the methodology lacks of a clear logical structure; for example, some of the tasks included in the section 2.1 "survey planning" (e.g. gathering information and pictures on damage location on the media and social networks) are also included in the preliminary considerations discussed in the section methodology.

Reply: according to this and the next comment, and also to Referee #1, we have improved the structure merging "Survey planning" and "previous considerations" sections into a new one titled "Pre in-situ survey tasks", to strength the chronological order of the tasks to carry out to apply the proposed methodology.

(3) Likewise, the section called "previous considerations" seems more related to preliminary considerations. With respect to this point, I think that a flowchart of the methodology, showing its steps in a logical order could support both possible users and readers.

Reply: thanks for the recommendation respect to adding a flowchart. We have presented it in Section 2 (Methodology), as an overview of the proposed methodology. Following, we attached the flowchart proposal.

[Figure]

Figure. Flow diagram of the structure and application of the proposed methodology to carry out strong-convective winds fieldwork damage assessment.

(4) Moreover, the proposed methodology is not a systematic or a standardised one: only some indications of which could be the different aspects to be surveyed are provided (see e.g. section 2.2.2) without a systematic and standardised procedure for their survey and collection (e.g. by means of pre-defined questions in a form). The only "step" that, in some way, is standardised is the witness enquiries (section 2.2.4), for which predefined questions are provided. The lack of standardisation is a big limit towards the objective of creating homogenous databases, given that the parameters/aspects to be surveyed, the way they must be surveyed/measured, and the possible values assumed by each of them is a subjective choice of the surveyor.

Reply: Thanks for your comment. We agree that the methodology can be better explained. To clarify and standardize the methodology, we have proposed to include two new tables. In the first one (Table 1 in the new version) there are listed all the variables which should be measured for each damaged element during the in-situ damage survey and the typical uncertainty:

| Variable | Uncertainty | Comments |
|---|---|---|
| Latitude | $\pm\ 1\cdot10^{-4}$ deg. | Measured with GPS camera. |
| Longitude | $\pm\ 1\cdot10^{-4}$ deg. | Measured with GPS camera. |
| Damage Indicator (DI) | --- | Determined during the post in-situ damage survey using intensity-rating scales as EF-scale. |
| Degree of Damage (DoD) | --- | Determined during the post in-situ damage survey using intensity-rating scales as EF-scale. |
| Previous weakness | --- | Description of deficiencies that can increase the vulnerability of elements to strong winds. |
| Dragged distance object | $\pm\ 1$ m | Distance between the final position and the origin of an object displaced by the wind. Measured with a tape measure or GIS tools. |
| Dragged direction object | $\pm\ 5$ deg. | Direction of the displacement. Measured with a compass or GIS tools. |
| Weight of wind-borne debris | $< 10\%$ | Weight of an object of interest moved by the wind. In case of small objects, measured with a balance if possible. |
| Fallen tree direction | $\pm\ 5$ deg. | In case of uprooted trees. Measured with a compass. |
| Trunk diameter | $\pm\ 5$ cm | In case of snapped trees. The trunk perimeter is measured with a tape measure and then the diameter can be calculated. |

And in the second (Table 3) it is presented the seven location types with each description and data that should be included:

| Location reference | Description | Data |
|---|---|---|
| L1 | Damage in trees with fall direction | Latitude, longitude, DI-DoD, previous weaknesses, fall direction |
| L2 | Damage in trees without fall direction | Latitude, longitude, DI-DoD, previous weaknesses, trunk diameter (if snapped tree) |
| L3 | Damage in man-made structures | Latitude, longitude, DI-DoD, previous weaknesses |
| L4 | AWS location | Latitude, longitude, data (maximum wind gust, direction of maximum wind gust and hour) |
| L5 | Witness location | Latitude, longitude of the witness location at the moment of the meteorological event and a brief description of his experience |
| L6 | Image of the phenomenon | Latitude, longitude from the point where image was recorded and orientation |
| L7 | Wind-borne debris | Latitude, longitude, distance and direction of the displacement, size and weight of the object if measured |

According to De Groeve *et al.* (2014) we have proposed some changes in final deliverables to standardize them including elements from Table 2 of the above mentioned study, showed in pages 34 and 35 (see Section 2.3 in the new submitted version and in the Supplementary material).

We have also attached the final deliverables of a case study to show explicitly how collected data is organized and presented.

(5) Objective of the survey. The objective of the survey is not really clear. Is it reproducing the damage scenario? Is it identifying the kind of event for insurance purposes?

Reply: the objective of the survey is reproducing the damage scenario to study the event from a meteorological point of view. With the gathered data: (1) the type of strong-convective wind phenomenon is identified, (2) the damage swath is characterized (it is determined the length and the width of the track) and (3) the intensity of the phenomenon is also estimated. In the particular case of Spain, as it is mentioned in the paper, the reinsurance public company also needs to know which kind of phenomenon caused damage, but the main interest of carrying out field works is characterizing events to build up a robust database about this kind of natural hazard. We have added this information in the introduction to clarify it.

(6) Given the effort requires by on-field surveys, I think that the multi-usability of collected data should be pursued (see references below). For example, what about the amount of damage data collected? Are they used only to characterise the hazard? This arises also the question of whether the products (deliverables) identified in the paper are suitable for multi-purposes uses of data Case studies.

Reply: all the collected data is used to characterise strong-convective winds (i.e. phenomenon type, intensity, damage path length and width) with the main aim of building up a robust and homogeneous database of this kind of meteorological phenomena. Moreover, as it is mentioned in De Groeve *et al.* (2013) and in De

Groeve *et al.* (2014), data gathered in a field work is also useful to further analyse the exposition and vulnerability of damaged man-made structures, and it is also possible to study the impact of strong convective wind phenomena in an area. In addition, all this information can also be used to enhance or compliment wind intensity rating scales (as presented in Mahieu and Wesolek, 2016), even to create a new one as it is proposed in Groenemeijer *et al.* (2019). These comments have been included in the corrected manuscript.

(7) Case studies do not supply examples of how to implement the methodology but simply describe the events and the scenarios resulting from the survey. I think this is due to the lack of standardised tools for the implementation of the methodology previously commented.

Reply: the main objective of analysing these two case studies was to explain in practical cases which handicaps we found and to show the final deliverables, similarly as it is done usually in meteorological case studies. Attending to Referee #2, we have attached the three deliverables from the 15 October 2018 Malgrat de Mar-Massanes tornado as supplementary material. Thus, it shows explicitly how all the data gathered during a strong-convective wind damage survey carried out following the proposed methodology is organized and presented in a practical way.

References

[revised manuscript text omitted]

---

## Referee Report (RR1)

Journal: NHESS
Title: **A methodology to conduct wind damage field surveys for high impact weather events from convective origin**
Author(s): Rodríguez et al.
MS No.: nhess-2019-294
MS Type: Research Article
**Iteration: Second Review**

I really appreciate the efforts made by authors; the manuscript improved with respect to the previous version. Especially, I think that the explanation of the methodology in subsequent steps (i.e. pre in-situ survey tasks, in-situ survey tasks, and post in-situ survey tasks) really supports its understanding, as well as the inclusion of a flow chart. However, the paper still suffers from some criticalities, which prevent its publication in the present form. In the following, such criticalities are explained in detail with also some specific comments. However, I reserve to work in deeper on minor specific comments, if/when a further version of the manuscript will be submitted.

**Main criticalities**

*Objective of the survey.* The objective of the survey is still not really clear or, at least, is not univocally defined. If authors state that the objective of the study is to propose "a methodology to conduct damage survey" or "to represent the damage scenario", I expect that the main output of the procedure is a damage assessment/map/report. This is not the case here where damage is used as proxy to estimate hazard (i.e. wind) features, along with other type of data (e.g. satellite imagery, data from Doppler radar, lightning detection systems, AWS). This is clearly explained in conclusions: i.e. "The purpose of the presented methodology is to provide a systematic and easily-reproducible methodology to carry out strong-convective wind event *(damage, I would delete)* surveys, mainly based on gathering geolocated information about damaged man-made structures and vegetation, with the final aim of representing the damage scenario to study the event from a meteorological point of view", but it is ambiguously described in the whole manuscript. I recommend authors to check the manuscript in this regard, and to avoid the use of "wind damage survey", in favour of a more-general "wind field survey".

*Methodology.* The organisation of the procedure in three steps supports readers in its understanding; the same could be ideally state for the flow chart. Unfortunately, the flow chart presently does not reproduce the procedure, which may hamper paper readability. As a first issue, the flow chart does not report many aspects of the procedure described in the text, which should be instead included. In particular, according to the text, step 1 (i.e. pre in-situ tasks) is not limited to (geo)locating damage/affected area on the bases of damage or developed funnel cloud reports (as presently described in Figure 2) but it also includes:
- Preliminary gathering of information about damage location and images available on the media and social networks
- Contacting emergency services and local authorities to verify if they recorded detailed damage data
- Analysing satellite and weather radar imagery to estimate the approximate timing of the event and the movement of the convective parent storm that may have produced the phenomenon
- checking the wind climatology of the studied area
- verifying if the studied area has been affected recently by another damaging windstorm or by a heavy snowfall which may have produced widespread damage in forest

Likewise, according to the text, final deliverables are not created only on the bases of surveyed damage data (as presently indicated in the flow chart) but also considering meteorological remote-sensing data (collected ex-post the in-situ survey, according to the text) like satellite imagery, data from Doppler radar, lightning detection systems and AWS. On the opposite, the flow chart identifies collection of AWS data, information on previous events, etc. as part of step 2 (i.e. in-situ survey), differently than what declared in the text. A second issue relates to the aspects presently included in the flow chart. In particular: (1) which process/action must be performed in the rectangular box "Only developed funnel cloud report available" in

step 1? (2) Question in the decision box in step 3 is not correctly formulated. According to Figure 2, if one is in case 1, no deliverables are produced, but this is not the case. I strongly suggest authors to redraft the working flow being coherent with the text.

*Standardisation.* Authors introduced Table 1 and Table 3 in order to better explain the "standardization" of the methodology; these certainly help. However, I think that some aspects of the methodology are still not standardised. For instance, is there any pre-defined form to be used in field survey? e.g. by reporting the DI-DoD for each affected elements, data to be surveyed and format, etc.; this is critical to gain objective data. In particular, the implementation of the EF-scale for non-expert readers is problematic, for the full comprehension of the procedure. I strongly recommend authors to include a brief explanation of the procedure in the text or a full explanation as supplementary material. At last, how measure uncertainty in table 1 has been defined should be documented in the paper.

*Discussion* This section is weak. First, only main difficulties in implementing the methodology are discussed, while no discussion is done on its strengths. For example, in section 2 authors state "The methodology to carry out damage surveys must be efficient, making possible to visit the affected area in the shortest possible time. It must be also easily reproducible and its results should be accurate". Does the proposed methodology satisfy these requirements? A discussion would be welcome. Likewise, in the introduction authors state that, besides supporting the interpretation of the phenomena from a meteorological point of view, the procedure can (ideally) support others objectives like: insurance compensation, increased understanding of exposure and vulnerability to winds, improved wind intensity rating scales. Does the procedure, in its present form, actually support this objectives? While I am quite sure that compensation from insurance companies can be supported, I am not sure that the present collected data can support the other two objectives. Also in this case, a discussion would be welcome. Second, regarding "difficulties", the explanation of how "the wind phenomenon type can be determined from forest damage patterns" is not a difficulty of the methodology but rather an example of how collected data can be used. This part should be put in a separate section, or removed from the paper (in fact, in my point of view, it is more in line with the supplement than with the main document).

**Specific comments**

Abstract "high-impact weather events such as floods or strong wind events" → I would not define floods as weather events, maybe weather related events?

Pg. 2 line 25 → please, check if "survey assessment" is properly used here

Pg. 2 line 28 "When these phenomena affect a sparsely populated area, or they occur in a low visibility environment due to night darkness or intense precipitation, there is usually a lack of direct witnesses and recorded images. In that case, the task of assessing the damage intensity and discriminating if it was caused by a tornado, a downburst or another type of convective winds can be very challenging" → this sentence is not linked with previous and following ones, consider removing

Pg. 2 line 33 "… the systematic elaboration of post-event forensic field surveys is still the standard way to evaluate the damage caused by these meteorological phenomena" → this concept has been already explained few lines above (line 25); please, pay attention to repetitions

Pg. 2 lines 39-46 → Discussion on motivation is quite poor. Please, elaborate more.

Pg. 2 lines 53-61 → I would move the discussion/presentation of further uses of collected data in the discussion section (see also main criticalities)

Pg. 3 line 77 → I think that Deliverable 1 is more than a "a summary of fieldwork" as it also includes data elaboration/interpretation (like characterisation of the event); please, consider rename/reclassify the deliverable

Pg. 3 line 84 "Smartphone or cameras with GPS image geolocation and orientation (azimuth pointing) capabilities provide essential data to carry out a fieldwork in order to geolocate damage, as mentioned previously" → this concept has been already explain few lines above (line 79); please, pay attention to repetitions

Pg. 5 line 143 "wind speed damage thresholds may be higher than in non-windy regions" → which thresholds are authors referring to? Not clear at this point of the manuscript

Pg. 9 line 264 "this part also must contain the start and end date and time (in UTC)" → inclusion of time data has been already discussed few lines above (line 262); please, pay attention to repetitions

Pg. 13 line 383 → please, check if "survey assessment data" is properly used here

Figure 2 → please, amend as suggested (see main criticalities)

Figure 3 → please, consider removing; no added information for text comprehensibility

Figure 5 → no reference is made in the text

Figure 9, caption → please, remove the typo "Map symbols indicate locations of damage in man-made structures (house icons) and fallen tree or damaged vegetation element (arrow or circle icons if no direction is available)".

---

## Referee Report (RR2)

Journal: NHESS
Title: **A methodology to conduct wind damage field surveys for high impact weather events from convective origin**
Author(s): Rodríguez et al.
MS No.: nhess-2019-294
MS Type: Research Article
**Iteration: Third Review**

I thank the authors for the further efforts made to develop the manuscript. I think it has improved a lot, with respect to the first and previous version, and is now suitable for publication in NHESS, after some minor revision to the text and a revision of section 1.

Indeed, I think that this section still represents a weakness of the manuscript, being confused, bad structured and with many repetitions, probably because of authors willingness to address all the comments made by reviewers on the need to better explain the objective of the paper with respect to the current state of art. I strongly encourage the authors to review this section; particularly, lines from 45 to 60 are really bad connected with the other contents and should be moved to a dedicated section or sub-section.

Finally, I think authors should put more efforts to justify uncertainty bands in table 1. The given answer "We may did not expressed in the exact way what these errors mean" is not acceptable.

**Specific comments**

Pg. 1 line 32 "The systematic elaboration of post-event forensic in-situ field surveys" → consider rephrasing; I guess in-situ and field are synonymies

Pg. 4 line 113 " Secondly, the in-situ fieldwork tasks, which include direct gathering of man-made structure and vegetation damage information, and also direct witness experiences"→ please, check. I think the verb is missing

Pg. 4 line 114 "Finally, post in-situ damage assessment tasks, which involve ordering and organising all the information collected into three deliverables "→ please, check. I think the verb is missing

Pg. 5 line 129 "i.e., extended at least 50% the distance between the cloud base and surface" → what is it? A feature of available data? A requirement? Not clear, please specify

Pg. 5 line 146-149 → considering moving this before line 129 to be coherent with Figure 2

Pg. 7 line 222 "As mentioned in Sect. 1 the maximum wind field (direction and intensity) associated with a strong-convective wind event can be approximately derived from the fallen trees pattern" → this discussion is not present anymore in section 1, please correct

Pg. 10 line 310 "together with a brief explanation from the initial information gathered before starting the damage survey" → replace "from" with "of"

Pg. 13 line 393 "to perform this kind of studies it may be necessary more detailed information" → for example?

Pg. 13 line 403 "This information is extremely valuable to analyse the impact of tornadoes in different scenarios and future projections" → please, expand this concept more, e.g. with some examples

Figure 2 → the location of the process " Contact with local authorities and emergency services" is not coherent with the main text, please amend the figure or the text (see previous comment)

Figure 2 → I would change the description of the starting point with "detection of an event"

Table 2, caption "Dragged distance, direction and weight of wind-borne debris should be measured." → please, indicate when these actions must be performed to be coherent with the remaining description

---

## Author Response (AR2)

**General comments**

1. The aim of the study needs to be stated more clearly.

Thanks for this comment. We have rewritten both the objective of the paper and the main aim of the methodology in the Introduction, also according to Anonymous Referee #3, as follows: "The objective of this paper is to propose a methodology to conduct in-situ damage surveys of strong wind events from convective origin. It can contribute to improve the detection, mapping and characterization of wind damage in a homogeneous way, which is important to better describe specific meteorological phenomena, with the particularities associated with damage from convective local storms. Therefore, the main goal of the proposed methodology is to gather as much geo-referenced information as possible about relevant damaged elements (i.e. man-made structures and vegetation) to reproduce the damage scenario and complement it with other available data (witness enquiries, Automatic Weather Station -AWS- located close to the affected area, and remote-sensing data) to analyse strong-convective winds phenomena from a meteorological point of view.".

2. The methodology is based on previous studies. The studies are just cited. There is no review of the previous literature to show how the methodology described in the current study compares with the previous methodologies.

According to this comment, we have added a new paragraph in Section 1 (Introduction) exposing the state-of-the-art about in-situ damage survey methodologies: "McDonald and Marshall (1984) and Bunting and Smith (1993) provided guidelines to carry out strong-convective wind damage surveys. There, the process of mapping data by locating images taken during the fieldwork was challenging due to non-digital cameras and the absence of Global Navigation Satellite System incorporated on these devices. Both documents recommended to complement surface observations with aerial images if available and, in the second one, it was also explained how to treat with direct witnesses and propounded to ask them for specific information about the event and damage. On the other hand, the analysis of historical events such as Gayà (2007) and Holzer et al. (2018) showed the utility of press references and in-situ images taken by witnesses on reconstructing tornado damage paths. They pointed out the necessity of geo-referencing the locations where photos were taken and the damaged elements, using GIS tools and triangulation methods. Furthermore, Holzer et al. (2018) provided useful indications for current fieldworks, such as visiting affected areas as soon as possible and also to provide an estimation of the wind intensity for each damaged element given by the pair Damage Indicator-Degree of Damage (DI-DoD) from the EF-scale (WSEC, 2006), similarly to other authors such as Burgess et al. (2014). During the first decade of the current century, the use of GPS receivers or similar systems was extended on in-situ damage assessments to geolocate gathered data, as discussed in Edwards et al. (2013). Moreover, the use of aerial imagery from helicopter (e.g., Fujita, 1981; Bech et al., 2009) and high-resolution satellites (e.g., Molthan et al., 2014; Chernokulsky and Shikhov, 2018) has also been frequent to analyse damage swaths. Recently, drones have been raised as a new device which might be useful to, at least, complement surface surveys (Bai et al., 2017).".

**Minor comments**

line 2: Replace „extension" with „extent".

       Done.

lines 4–5: „[...] (i.e., tornado, downburst, straight-line winds)".

       Done.

line 5: „[...] between 2004–2018".

       Done.

line 6: „systematic collection of images"? This is also repeated toward the end of the phrase.

       Here we would like to emphasise that pictures of damage are taken during the fieldwork by surveyors, whereas images of the phenomenon are taken by direct witnesses and gathered by the damage survey team. We have rewritten the sentence as "The methodology includes the collection of pictures and records of damage on man-made structures and on vegetation during the affected area in-situ visit, as well as of available Automatic Weather Station data, witness reports and images of the phenomenon, such as funnel cloud pictures, taken by casual observers.".

line 6: The word "systematic" is used too many times in the manuscript (developing a methodology already means that will be applied systematically).

       Thanks for this observation. We have removed the word "systematic" here and also from several sentences along the manuscript.

line 8: „ recorded during the damage field survey".

       We have rewritten this sentence as follows: "To synthesize the gathered data, three final deliverables are proposed:".

line 18: „surface wind of convective origin".

       Done.

line 18–19: „(i.e., tornadoes, downbursts, straight-line winds)".

       Done.

line 19: „or even fatalities".

       Done.

line 22: „[...] devoted to the study of these phenomena".

       Done.

line 23: „(e.g., Strader et al., 2015)".

       Done.

line 24: Replace „see for example" with „e.g.".

       Done.

line 30: Replace "discriminating" with "identifying".

We have removed this sentence.

line 36: "allows a detailed characterization".

We have rewritten this sentence as: "wind damage fieldworks contribute to a better characterization of the affected area, making possible to estimate the length and width of the damage swath".

lines 39–44: This paragraph needs to be rewritten and reduced for clarity; one way will be to combine this paragraph with the next one and thus to state more clearly the aim of the study.

We have rewritten for more clarity and developed the motivation part, according to this comment and also to Anonymous Referee #3: "Currently, fieldworks are usually performed to assess damage of specific strong-convective wind events, but rarely to analyse in detail most of the reported cases. The timing and economical costs, especially when helicopter flights are used, prevents to carry out fieldworks frequently (Edwards, 2020), particularly out of the USA. Therefore, there is a need for a methodology to conduct wind damage field surveys for high impact weather events of convective origin easily reproducible anywhere, which should be efficient and optimize time and economic resources to allow analysing as much reported events as possible. This methodology may contribute to the homogeneity of severe weather reports databases, improving the knowledge of spatial-temporal distribution and characteristics of tornadoes, downbursts and straight-line winds.".

line 44: No mention of the study area so far.

We have added a sentence briefly presenting the area of study when it is mentioned for the first time (now, in the penultimate paragraph of the Introduction): "All the analysed events have been recorded in Spain (south-western Europe), which includes a vast majority of the Iberian Peninsula and also the Balearic and Canary Islands. Nevertheless, most of these fieldworks have been carried out in Catalonia and Andalusia regions (Fig. 1), where high-densely populated areas are frequently affected by tornadoes (Bech et al., 2007, 2011; Mateo et al., 2009; Gayà et al., 2011; Sánchez-Laulhé, 2013; Riesco et al., 2015).".

line 47–57: state more clearly the aim of the study.

As it has been explained before, we have rewritten the objective of the study in the Introduction to make it more clear.

lines 72–74: I think this example should be included in the manuscript as a section.

Thanks for this recommendation. In the first submitted version of the manuscript it was proposed a section where this event was analysed throughout results obtained from using the methodology (see https://www.nat-hazards-earth-syst-sci-discuss.net/nhess-2019-294/nhess-2019-294.pdf). Nevertheless, a reviewer proposed to attached it as Supplementary material. That was the motivation to remove it from the main text.

line 76: describe "previous related work".

Here we have mentioned the cited articles which we refer to.

line 85: Remove "to geolocate damage, as mentioned previously".

Done.

lines 84–92: Maybe this information can be summarized in a table.

We have summarized the devices which are required to perform a strong-convective wind damage survey in a table (now, Table 1), following partially this comment. Other recommendations have been remained in the main text, which has been rewritten as follows: "In Table 1 are summarized the main devices needed to carry out fieldworks throughout the proposed methodology. Moreover, as indicated in Bunting and Smith (1993) and Gayà (2018), water, food, comfortable footwear, rain jacket, spare clothes and a mobile phone spare battery are recommended, because affected areas may be far away from inhabited locations. As surveyor displacements longer than a few kilometres can be required, a well-equipped, preferably all terrain, car is necessary to save time between points-of-damage analysis. Nevertheless, difficult access areas may be found along the track, because of muddy roads and fallen trees or simply because of the absence of roads. Especially in these cases, and also to study in detail damaged areas, walking is the basic way to perform the field survey.".

line 97: "[…] in three stages (Fig. 2)".

Done.

lines 97–101: Add references to subsections.

Thanks. We completely agree that mentioning subsections will make more clear the article structure and it would be easier for the readers to find the information.

line 102: "To properly […]".

Done.

line 106–107: "[…] during the survey (Fig. 3)".

We have removed this Figure according to Anonymous Referee #3.

line 107: "Thus, to optimize time and resources, a detailed […]".

Done.

line 110: "[…] reports nowadays should be collected".

Done.

line 115–116: Why a "developed funnel cloud"? Why not a tornado? Funnel cloud may not be associated with damages. This aspect should be argued more strongly.

This is an important point, and we thank you for commenting this, because indicates that it was not clear on the text. Tornado definition implies that the vortex is in contact with the surface, causing damage if vulnerable and exposed elements are present. When we mention a developed funnel cloud instead of tornado, we are assuming that the vortex may have not been in contact with the surface and may have not produced damage. In that case (Case 2 in Fig. 2) carrying out an in-situ visit of the possible affected area would confirm if the vortex was in contact with surface (tornado) or not (only

funnel cloud without tornado). We have modified these paragraphs, also according to Anonymous Referee #3 (new line 129 to new line 145).

line 115: replace "(see Fig. 4)" with "(Fig. 4)".

Done.

line 117: "by contacting their authors".

Done.

line 119: "if damage reports are not available".

Done.

line 119: Define "developed funnel cloud".

We have included the definition of developed funnel cloud as "extended at least 50% the distance between the cloud base and surface".

line 120–121: Provide details on how to locate these events based on radar and lightning data, or at least include references.

We have added the comment "comparing radar images and the location of precipitation features observed in photos of the event respect to the funnel cloud" and Wakimoto and Lew (1993), Wakimoto and Liu (1998) and Zehnder (2007) as references, where photogrammetry and radar observations are combined.

line 121: This sentence should be rewritten for clarity.

We have rewritten this sentence and merged with the next one: "Then, from the triangulation of those pictures of the funnel cloud it is possible to preliminarily identify a possibly affected area, which can be more precisely delimited when the number of photos or videos from different perspectives is high (Rasmussen et al., 2003).".

line 127–130: Is this not delaying the damage survey?

The information provided by emergency services could be very useful for complementing the data gathered during in-situ damage survey. Contacting with them does not prevent surveyors to start wind damage assessment although no information is received yet, because emergency services data would be used during post in-situ survey tasks. Accordingly, we have modified the second sentence of the paragraph to clarify this fact: "This kind of information may be crucial for post in-situ damage study because clearing services might start arrangement tasks before the in-situ visit is started".

line 134: Define "indirect eyewitnesses".

When we mentioned indirect eyewitnesses here we wanted to refer to those people who was not in the affected area when damaging-winds took place but they know about damage. Because it could be confusing, we have removed it.

line 142: "In this case".

Done.

line 145: Replace "relaxing" with "reducing".

We have modified this sentence (see new line 167).

line 154: "of the damage".

Done.

line 155: "to recover quickly".

Done.

line 157: "(Fig. 2)".

Done.

line 159: "[…] interviews with eyewitnesses".

Done.

line 176: "[…] to be performed during".

Done.

line 197: "[…] might not be".

Done.

line 208: "(Fig. 6a)".

Done.

line 218: Replace "To solve that" with "In this case".

Done.

line 222: Replace "reported" with "developed".

Done.

line 230: "[…] in their own words".

Done.

line 238: This sentence is about photo/videos or eyewitnesses?

This sentence is about images of the event. We have rewritten the paragraph to make it clearer: "A brief and concise inquiry, with specific questions but allowing open answers that may unveil relevant information, is proposed. Recommended questions are shown in Table 3. Moreover, in some occasions a direct witness may have taken photos or videos of the phenomenon that can be helpful for the study. When pictures are available, they should be treated as described in Sect. 2.1.".

line 262–282: Add these as bullet points.

Done.

line 328: Remove extra ")".

Done.

lines 340–342: I do not understand this sentence.

We have modified this sentence as follows: "Despite real wind damage patterns can be very complex due to its interaction with topography or with other nearby events (Forbes and Wakimoto, 1983; Cannon et al., 2016), theoretical-idealized damage swath patterns of both tornado and downburst wind fields can be compared with observed damage patterns in order to look for similarities to assess their possible origin.".

line 365: "[…] how debris".

We have changed to "the way debris is spread" to avoid repeating "how", only few words later.

lines 368–370: The authors should comment on the results, this is just a description of the figures.

These two sentences were motivated to introduce the figure about the following discussion was centred. Nevertheless, according to this comment we have rewritten all the paragraph: "As a real example, the case shown in Fig. A2 presents fallen poplar trees following a convergence pattern: in the right-half side from the damage swath, trees are blown down to the west, whereas in the left-half side they are uprooted to the north. Comparing this real case and idealised cases (Fig. A1), this damage pattern matches well the damage swath caused by a vortex with $G = 4$ and $\alpha = 45°$ (Fig. A1a). This fact along with other evidences confirm the hypothesis that damage was caused by a tornado, as presented in the Supplementary material. Moreover, it is remarkable that these vortex characteristics are also coherent with the damage rated as the lower EF1 bound and the mean translational velocity of 12 m s$^{-1}$, estimated using radar data from the Meteorological Service of Catalonia (not shown).".

References: Some of the abbreviations for the journals are not in the correct format (e.g., B. Am, Meteorolog. Soc. should be Bull. Amer. Meteor. Soc.).

Thanks for noticing this issue. We have carefully looked at other papers published in Natural Hazards and Earth Systems Science and we have checked that the abbreviation for BAMS used there is "B. Am. Meteorol. Soc.", which we used to replace "B. Am. Meteorolog. Soc.".  We have also revised the rest of references.

Figure 1: This figure is incomplete, see the inset map.

Thanks for pointing out this misunderstanding. We have changed the shaded colour of surrounding countries -and labelled them- to remark the area of study (Spain). The name of the two main Spanish archipelagos (Balearic and Canary Islands) have also been indicated. We hope that these changes may help to clearly locate the region of interest. Moreover, we have added the following sentence in figure caption to indicate the meaning of black and grey contours in the map: "Black contours delimitate regions and grey lines, provinces.".

[Figure]

**Anonymous Referee #3**

I really appreciate the efforts made by authors; the manuscript improved with respect to the previous version. Especially, I think that the explanation of the methodology in subsequent steps (i.e. pre in-situ survey tasks, in-situ survey tasks, and post in-situ survey tasks) really supports its understanding, as well as the inclusion of a flow chart. However, the paper still suffers from some criticalities, which prevent its publication in the present form. In the following, such criticalities are explained in detail with also some specific comments. However, I reserve to work in deeper on minor specific comments, if/when a further version of the manuscript will be submitted.

**Main criticalities**

*Objective of the survey*. The objective of the survey is still not really clear or, at least, is not univocally defined. If authors state that the objective of the study is to propose "a methodology to conduct damage survey" or "to represent the damage scenario", I expect that the main output of the procedure is a damage assessment/map/report. This is not the case here where damage is used as proxy to estimate hazard (i.e. wind) features, along with other type of data (e.g. satellite imagery, data from Doppler radar, lightning detection systems, AWS). This is clearly explained in conclusions: i.e. "The purpose of the presented methodology is to provide a systematic and easily-reproducible methodology to carry out strong-convective wind event (damage, I would delete) surveys, mainly based on gathering geolocated information about damaged man-made structures and vegetation, with the final aim of representing the damage scenario to study the event from a meteorological point of view", but it is ambiguously described in the whole manuscript.

> Thank you for this detailed and constructive comment. In the introduction it was specified, separately, the main objective of the paper (to provide a methodology to perform damage surveys) and of the methodology (to represent the damage scenario to study strong-convective winds phenomena from a meteorological point of view, which was very similarly exposed in Conclusions). Nevertheless, according to this comment, we have rewritten more clearly that the aim of the methodology it is not only gathering data of damaged elements, but also witness experiences, and AWS and remote-sensing data for a further analysis of strong-convective wind events. Therefore, in the new submitted version it is stated in the Introduction as: "Therefore, the main goal of the proposed methodology is to gather as much geo-referenced information as possible about relevant damaged elements (i.e. man-made structures and vegetation) to reproduce the damage scenario and complement it with other available data (witness enquiries, Automatic Weather Station -AWS- located close to the affected area, and remote-sensing data) to analyse strong-convective winds phenomena from a meteorological point of view."; and in Conclusions as: "The purpose of this article is to provide an easily reproducible methodology to carry out surface strong-convective wind event damage surveys, which optimize time and economic resources. It is mainly based on collecting geolocated information about damaged man-made structures and vegetation, with the final aim of representing the damage scenario to study the event from a meteorological point of view. Complementary data from AWS close to the affected area and witness reports should also be gathered if available, and remote-sensing data should be used to get a deeper understanding of the convective storm event.".

I recommend authors to check the manuscript in this regard, and to avoid the use of "wind damage survey", in favour of a more-general "wind field survey".

We agree that the term "wind field survey" would be more general. However, we believe that in order to be consistent with the terminology used in the related literature it is important to keep the term "wind damage survey" for two reasons. Firstly, the survey methodology we propose is motivated by the occurrence of damaging winds of convective origin. Secondly, the estimation of wind conditions (intensity and direction wind field patterns, type of convective phenomenon) is done through recognition of wind effects upon damaged elements and therefore the analysis is implicitly considered only, again, for damaging winds. For these two reasons we think that the use of the term "wind damage survey" is more appropriate and precise than "wind field survey".

*Methodology*. The organisation of the procedure in three steps supports readers in its understanding; the same could be ideally state for the flow chart. Unfortunately, the flow chart presently does not reproduce the procedure, which may hamper paper readability. As a first issue, the flow chart does not report many aspects of the procedure described in the text, which should be instead included. In particular, according to the text, step 1 (i.e. pre in-situ tasks) is not limited to (geo)locating damage/affected area on the bases of damage or developed funnel cloud reports (as presently described in Figure 2) but it also includes:

- Preliminary gathering of information about damage location and images available on the media and social networks

- Contacting emergency services and local authorities to verify if they recorded detailed damage data

- Analysing satellite and weather radar imagery to estimate the approximate timing of the event and the movement of the convective parent storm that may have produced the phenomenon

- checking the wind climatology of the studied area

- verifying if the studied area has been affected recently by another damaging windstorm or by a heavy snowfall which may have produced widespread damage in forest

Likewise, according to the text, final deliverables are not created only on the bases of surveyed damage data (as presently indicated in the flow chart) but also considering meteorological remote-sensing data (collected ex-post the in-situ survey, according to the text) like satellite imagery, data from Doppler radar, lightning detection systems and AWS. On the opposite, the flow chart identifies collection of AWS data, information on previous events, etc. as part of step 2 (i.e. in-situ survey), differently than what declared in the text.

We are very thankful for the constructive recommendations and comments given. We have rebuilt the flow diagram following them and being coherent with the main text. It has been added two process boxes into the first step -after the report receiving-, which are "Search information/images about the event on media and social networks" and, following, "Contact with local authorities and emergency services". Moreover, before starting the second step, one more process box has been incorporated, where it is mentioned the necessity of analysing meteorological data to define the timing of the event, the translational direction of the convective cell, the wind climatology of the

affected area and if there has been any recent damaging event in the region of study. Note that in the main text (Sect. 2.2), when in-situ survey tasks are briefly presented, it is explained that, although during pre in-situ survey tasks it is recommended to search for AWS data in the area, it is also suggested to search other AWS during the fieldwork which were previously unknown. Moreover, it is pointed out the valuable information that outdoor security cameras may provide. Finally, the third step has been also restructured to clarify how to proceed after performing the fieldwork.

A second issue relates to the aspects presently included in the flow chart. In particular:

(1) which process/action must be performed in the rectangular box "Only developed funnel cloud report available" in step 1?

In fact, it is not an action; it is a result (that only developed funnel cloud is reported, without known damage). We have corrected the error by changing the process (rectangular) box for a data box.

(2) Question in the decision box in step 3 is not correctly formulated. According to Figure 2, if one is in case 1, no deliverables are produced, but this is not the case.

Thanks for this detailed analysis. We have reorganized the flow chart to show specifically that in case 1 the set of three deliverables are generated, according to the text.

I strongly suggest authors to redraft the working flow being coherent with the text.

We attach below the new version of the flow chart.

[Figure]

*Standardisation*. Authors introduced Table 1 and Table 3 in order to better explain the "standardization" of the methodology; these certainly help. However, I think that some aspects of the methodology are still not standardised. For instance, is there any pre-defined form to be used in field survey? e.g. by reporting the DI-DoD for each affected elements, data to be surveyed and format, etc.; this is critical to gain objective data. In particular, the implementation of the EF-scale for non-expert readers is problematic, for the full comprehension of the

procedure. I strongly recommend authors to include a brief explanation of the procedure in the text or a full explanation as supplementary material.

All the data to be surveyed and gathered is indicated on the manuscript (i.e., latitude; longitude; previous weakness -and images- of each damaged element; fall of direction for uprooted tree; trunk perimeter for snapped tree; distance between the final position and the origin of an object displaced by the wind and direction; weight of an object of interest moved by the wind). There is not any pre-defined form to be used during the fieldwork because we have seen during damage assessments performed that there are other useful and easy ways to gather them, which is now explained in Subsection 2.2: "All the in-situ measurements are related to geo-referenced damaged elements. To reduce the time of registering data in order to proceed to the visit of other affected areas, it is proposed to take a photo from the measuring device showing clearly the data (Fig. 5), following the order proposed in Table 2 (from V5 to V10). After that, a photo of the damaged element should be taken, whose metadata already contains latitude and longitude. Therefore, surveyors can associate each measure to each geolocated element during the post in-situ tasks when organising the records gathered in the fieldwork.". We have also added a new figure (attached below, now Fig. 5) to better illustrate this procedure.

[Figure]

We agree that EF-scale application may be problematic for experts and non-experts (in Discussion section these challenges are explained). Nevertheless, the use of damage rating scales is the most common method to estimate the wind intensity. As indicated in the main text, the task of assigning a DI-DoD pair for each damaged element can be carried out during damage assessment, although we recommend doing it during the post in-situ damage survey analysis to optimize time. When surveyors have to carry on damage rating tasks, they consult specific documents such as WSEC (2006), a document to which readers are also referred (new line 195 to new line 198 and new line 205 to new line 209).

At last, how measure uncertainty in table 1 has been defined should be documented in the paper.

Thanks for this comment. We may did not expressed in the exact way what these errors mean. They are the maximum uncertainty recommended for each measure (in the new submitted version we have specified it on Table caption). We consider that data with greater errors may not be useful to proceed on the analysis (i.e., increasing error in

coordinates could provide deficiencies on length/width damage path estimation; in fall direction of trees would alter the estimated wind field and make more difficult to know which phenomenon took place, etc.). We have added a new paragraph commenting these considerations in the main text (Subsection 2.2; new line 190 to new line 192).

*Discussion* This section is weak. First, only main difficulties in implementing the methodology are discussed, while no discussion is done on its strengths. For example, in section 2 authors state "The methodology to carry out damage surveys must be efficient, making possible to visit the affected area in the shortest possible time. It must be also easily reproducible and its results should be accurate". Does the proposed methodology satisfy these requirements? A discussion would be welcome.

Likewise, in the introduction authors state that, besides supporting the interpretation of the phenomena from a meteorological point of view, the procedure can (ideally) support others objectives like: insurance compensation, increased understanding of exposure and vulnerability to winds, improved wind intensity rating scales. Does the procedure, in its present form, actually support this objectives? While I am quite sure that compensation from insurance companies can be supported, I am not sure that the present collected data can support the other two objectives. Also in this case, a discussion would be welcome.

> We appreciate this comment. In the new submitted version we have rewritten and reorganised this section, discussing the strengths and weaknesses of the methodology (as proposed) and the use that can be made from the gathered data (please, see Section 3 in the manuscript).

Second, regarding "difficulties", the explanation of how "the wind phenomenon type can be determined from forest damage patterns" is not a difficulty of the methodology but rather an example of how collected data can be used. This part should be put in a separate section, or removed from the paper (in fact, in my point of view, it is more in line with the supplement than with the main document).

> Thanks for the comment. We agree that the determination of the strong-convective wind phenomenon type is not a handicap of the presented methodology, but a difficulty for the interpretation of results from fieldwork. For this reason, we think that it is interesting for readers to show how to relate forest damage pattern with phenomenon type. Therefore, we propound to attach this subsection as an Appendix, with the aim of providing a guide for future strong-convective wind damage surveyors to analyse fieldwork data.

**Specific comments**

Abstract "high-impact weather events such as floods or strong wind events" → I would not define floods as weather events, maybe weather related events?

> We have made this change.

Pg. 2 line 25 → please, check if "survey assessment" is properly used here

> We have removed this sentence.

Pg. 2 line 28 "When these phenomena affect a sparsely populated area, or they occur in a low visibility environment due to night darkness or intense precipitation, there is usually a lack of direct witnesses and recorded images. In that case, the task of assessing the damage intensity and discriminating if it was caused by a tornado, a downburst or another type of convective winds can be very challenging" → this sentence is not linked with previous and following ones, consider removing.

According this comment and the following one, both paragraphs (line 25 to line 38 from the previous submitted version of the manuscript) have been merged and rewritten: "The systematic elaboration of post-event forensic in-situ field surveys is still the standard way to evaluate the damage caused by a strong-convective wind event (Marshall, 2002; Marshall, 2012; Zanini et al., 2017), despite the recent progress on assessing wind damage using remote sensing data such as high resolution radar observations (Wurman et al., 2013; Wakimoto et al., 2018). A detailed damage analysis from these meteorological phenomena allows to estimate the wind intensity using a wind damage scale such as the Fujita scale (F-scale, Fujita, 1981) or the Enhanced Fujita scale (EF-scale, WSEC, 2006). Similarly to field surveys of hailstorms (Farnell et al., 2009) or floods (Molinari et al., 2014; Li et al., 2018), wind damage fieldworks contribute to a better characterization of the affected area, making possible to estimate the length and width of the damage swath (e.g., Burgess et al., 2014; Meng and Yao, 2014; Bech et al., 2015). Moreover, in-situ damage survey are especially useful to determine which phenomenon took place when there is an absence of observations by analysing damage patterns on forest and how debris is spread (Hall and Brewer, 1959; Holland et al., 2006; Bech et al., 2009; Beck and Dotzek, 2010; Rhee and Lombardo, 2018). This information can be added to natural hazards databases such as the USA Storm Prediction Center Severe Weather Database (Verbout et al., 2006) or the European Severe Weather Database (Dotzek et al., 2009), making possible building up robust and homogeneous datasets.".

Pg. 2 line 33 "… the systematic elaboration of post-event forensic field surveys is still the standard way to evaluate the damage caused by these meteorological phenomena" → this concept has been already explained few lines above (line 25); please, pay attention to repetitions

Thanks for this comment. As it has been explained in the previous item, the paragraph from where this sentence was part of has been rewritten.

Pg. 2 lines 39-46 → Discussion on motivation is quite poor. Please, elaborate more.

We have rewritten this part, according to this comment and also to Anonymous Referee #1: "Currently, fieldworks are usually performed to assess damage of specific strong-convective wind events, but rarely to analyse in detail most of the reported cases. The timing and economical costs, especially when helicopter flights are used, prevents to carry out fieldworks frequently (Edwards, 2020), particularly out of the USA. Therefore, there is a need for a methodology to conduct wind damage field surveys for high impact weather events of convective origin easily reproducible anywhere, which should be efficient and optimize time and economic resources to allow analysing as much reported events as possible. This methodology may contribute to the homogeneity of severe weather reports databases, improving the knowledge of spatial-temporal distribution and characteristics of tornadoes, downbursts and straight-line winds.".

Pg. 2 lines 53-61 → I would move the discussion/presentation of further uses of collected data in the discussion section (see also main criticalities)

> We have moved this part to discussion section and expanded.

Pg. 3 line 77 → I think that Deliverable 1 is more than a "a summary of fieldwork" as it also includes data elaboration/interpretation (like characterisation of the event); please, consider rename/reclassify the deliverable

> We agree with this comment and, therefore, we propose to rename this deliverable as "text report of the event".

Pg. 3 line 84 "Smartphone or cameras with GPS image geolocation and orientation (azimuth pointing) capabilities provide essential data to carry out a fieldwork in order to geolocate damage, as mentioned previously" → this concept has been already explain few lines above (line 79); please, pay attention to repetitions

> According to Anonymous Refree #1 we have listed required devices on a table, so we have removed this sentence.

Pg. 5 line 143 "wind speed damage thresholds may be higher than in non-windy regions" → which thresholds are authors referring to? Not clear at this point of the manuscript

> In this sentence, we are referring, in general, to the wind speed over which an element can be damaged. We have rewritten as follows: "wind speed thresholds over which an element can be damaged may be higher than in non-windy regions.".

Pg. 9 line 264 "this part also must contain the start and end date and time (in UTC)" → inclusion of time data has been already discussed few lines above (line 262); please, pay attention to repetitions

> Thanks for this comment. We have removed the first time reference.

Pg. 13 line 383 → please, check if "survey assessment data" is properly used here

> We have changed "survey assessment data" for "in-situ damage survey data".

Figure 2 → please, amend as suggested (see main criticalities)

> Done.

Figure 3 → please, consider removing; no added information for text comprehensibility

> We have removed this Figure, according to this consideration.

Figure 5 → no reference is made in the text

> Figure 5 (now, Figure 4) was already cited in the main text in Section 2.1 (Line 138 of the previous version manuscript) when it was discussed the importance of gathering high-temporal resolution data from AWS located close to the affected area. In the new version we have emphasised on the typical pattern that wind velocity and direction presents in temporal series when a vortex or a downburst passes close to the instruments: "
[revised manuscript text omitted]

775

---

## Author Response (AR3)

I thank the authors for the further efforts made to develop the manuscript. I think it has improved a lot, with respect to the first and previous version, and is now suitable for publication in NHESS, after some minor revision to the text and a revision of section 1.

Indeed, I think that this section still represents a weakness of the manuscript, being confused, bad structured and with many repetitions, probably because of authors willingness to address all the comments made by reviewers on the need to better explain the objective of the paper with respect to the current state of art. I strongly encourage the authors to review this section; particularly, lines from 45 to 60 are really bad connected with the other contents and should be moved to a dedicated section or sub-section.

> Thank you for your new careful review. We have reorganised section 1 content into the introductory section and a new background section following your advice.

Finally, I think authors should put more efforts to justify uncertainty bands in table 1. The given answer "We may did not expressed in the exact way what these errors mean" is not acceptable.

> We apologize for the previous answer. We have rephrased and expanded the comments about this item with this text: "Note also that Table 2 lists maximum uncertainties recommended for each measure type reflecting possible maximum errors in the field survey measures as suggested in Beven et al. (2018). Particular uncertainty values listed in Table 2 are consistent with the resolution of data presented in several severe weather databases such as NOAA/SPC (2019), ESWD (Dotzek et al., 2009), KERAUNOS (2020), SINOBAS (Gutiérrez et al., 2015) and Gayà (2018), where damage path width is usually expressed with a resolution of $\pm 10$ m (i.e., damage location uncertainty must be smaller than $\pm 1 \times 10^{-4}$ deg.). Furthermore, uncertainties listed also take into account surveyors experience and the data resolution from previous studies – e.g., direction of fallen trees and wind-borne debris are typically presented with 5 deg. range (Bech et al., 2009, 2011, 2015).".

**Specific comments**

Pg. 1 line 32 "The systematic elaboration of post-event forensic in-situ field surveys" → consider rephrasing; I guess in-situ and field are synonymies.

> We have modified this sentence by removing "in-situ": "The systematic elaboration of post-event forensic field surveys".

Pg. 4 line 113 "Secondly, the in-situ fieldwork tasks, which include direct gathering of man-made structure and vegetation damage information, and also direct witness experiences" → please, check. I think the verb is missing.

> Thanks. In the new submitted version, the sentence is as follows: "Secondly, the in-situ fieldwork tasks, which include direct gathering of man-made structure and vegetation damage information, and also collection of direct witness experiences (Sect. 3.2).".

Pg. 4 line 114 "Finally, post in-situ damage assessment tasks, which involve ordering and organising all the information collected into three deliverables" → please, check. I think the verb is missing.

We have modified the sentence as: "Finally, post in-situ damage assessment tasks, which involve the organisation of all the information collected into three deliverables (a text report of the event, a geolocated information table and a data location map; Sect. 3.3).".

Pg. 5 line 129 "i.e., extended at least 50% the distance between the cloud base and surface" → what is it? A feature of available data? A requirement? Not clear, please specify.

It is a requirement. If the reported funnel cloud is extended less than 50% the distance between the cloud base and the surface and no damage is reported, it should be aware to perform an in-situ damage survey. We have rewritten the sentence to make this point clearer: "To consider performing an in-situ damage survey, the report must contain information about damage and/or a well-developed funnel cloud (i.e., a funnel cloud extending down below cloud base at least 50 % the distance between the cloud base and the ground level).".

Pg. 5 line 146-149 → considering moving this before line 129 to be coherent with Figure 2.

Done.

Pg. 7 line 222 "As mentioned in Sect. 1 the maximum wind field (direction and intensity) associated with a strong-convective wind event can be approximately derived from the fallen trees pattern" → this discussion is not present anymore in section 1, please correct.

Thanks for pointing out this item. We have replaced the Sect. 1 citation with the references Holland et al. (2006) and Bech et al. (2009).

Pg. 10 line 310 "together with a brief explanation from the initial information gathered before starting the damage survey" → replace "from" with "of".

Done.

Pg. 13 line 393 "to perform this kind of studies it may be necessary more detailed information" → for example?

This sentence has been removed in the corrected version of the manuscript.

Pg. 13 line 403 "This information is extremely valuable to analyse the impact of tornadoes in different scenarios and future projections" → please, expand this concept more, e.g. with some examples.

We have rephrased the sentence as follows: "This information can be very valuable to analyse the impact of tornadoes in future projected scenarios, for example modelling intensity damage areas in tornado paths using the data provided by fieldworks and assessing the possible consequences under different expected urban conditions (Ashley et al., 2014; Rosencrants and Ashley, 2015).".

Figure 2 → the location of the process "Contact with local authorities and emergency services" is not coherent with the main text, please amend the figure or the text (see previous comment).

We have modified the text according to the previous comment.

Figure 2 → I would change the description of the starting point with "detection of an event".

Thanks for the comment. We have done this change.

Table 2, caption "Dragged distance, direction and weight of wind-borne debris should be measured." → please, indicate when these actions must be performed to be coherent with the remaining description.

[revised manuscript text omitted]